# What Do Large Language Models Know About Opinions?

**Erfan Jahanparast, Zhiqing Hong & Serina Chang**
University of California, Berkeley
`erfanjp,zhiqing,serinac@berkeley.edu`

## Abstract

What large language models (LLMs) know about human opinions has important implications for aligning LLMs with human values, simulating humans with LLMs, and understanding what LLMs learn during training. While prior works have tested LLMs' knowledge of opinions via their next-token outputs, we present the first study to probe LLMs' *internal* knowledge of opinions, evaluating LLMs across 22 demographic groups on a wide range of topics. First, we show that LLMs' internal knowledge of opinions far exceeds what is revealed by their outputs, with a 52-66% improvement in alignment with the human answer distribution; this improvement is competitive with fine-tuning but nearly $300\times$ less computationally expensive. Second, we find that knowledge of opinions emerges rapidly in the middle layers of the LLM and identify the final unembeddings as the source of the discrepancy between internal knowledge and outputs. Third, using sparse autoencoders, we trace the knowledge of opinions in the LLM's residual stream back to attention heads, and we identify specific attention head features that selectively encode different demographic groups. Through steerability experiments, we show that manipulating these features causally alters the LLM's outputs, aligning them more or less closely with different groups. These findings open new avenues for building value-aligned and computationally efficient LLMs, with applications in survey research, social simulation, and human-centered AI. Our code is available at `https://github.com/schang-lab/llm-opinions`.

## 1 Introduction

Human opinions are diverse and complex, reflecting how people feel about topics ranging from social issues to healthcare to foreign relations (Chu et al., 2023; Hu et al., 2025). In this work, we seek to understand: *what do large language models (LLMs) know about human opinions?*

This question has several important implications. First, as LLMs increasingly interact with and act on behalf of humans, there is an urgent need for pluralistic AI systems that can serve people with diverse values and perspectives (Sorensen et al., 2024; Kirk et al., 2024; Feng et al., 2024). To achieve pluralistic alignment—whether aiming for responses that steer towards a particular group or cover a spectrum of opinions—LLMs first need to know what different people's opinions are on different topics. Second, there has been great interest from survey researchers in developing LLM "synthetic respondents" for public opinion surveys, which require LLMs to have a deep understanding of human opinions (Argyle et al., 2023; Liu et al., 2025; Rothschild et al., 2024). Third, investigating LLMs' knowledge of opinions helps us better understand what they learn during training; for example, prior work has shown that LLMs learn concepts such as truthfulness (Marks & Tegmark, 2024), space and time (Gurnee & Tegmark, 2024), and the liberal-conservative spectrum (Kim et al., 2025). In contrast to these relatively simpler concepts, which fall along one or two dimensions, human opinions are far more multidimensional. Yet it remains unknown how well LLMs have learned such complex concepts or how they are encoded.

Despite many recent efforts to evaluate LLMs' knowledge of opinions, they have all relied on LLMs' outputs, either using the next-token probabilities over answer options or sampled generations (Santurkar et al., 2023; Hwang et al., 2023; Chu et al., 2023; Durmus et al., 2023; Feng et al., 2024; Moon

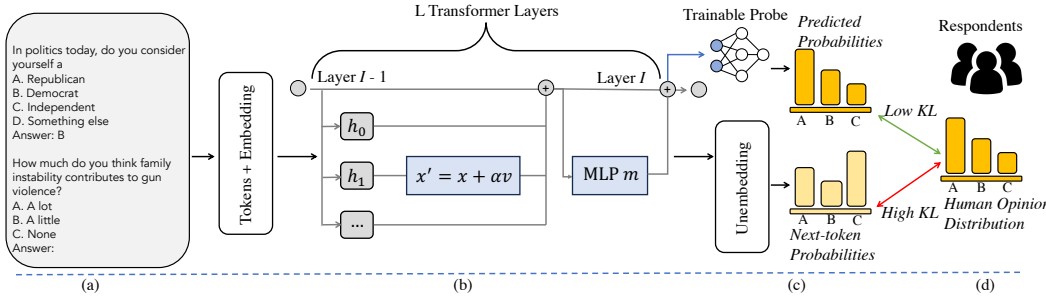

Figure 1: Illustration of our framework. (a) is an example of public opinion-based prompts. (b) is the transformer layers of LLMs. (c) is two ways to extract LLMs' knowledge, i.e., trainable probes (***probing***) and next-token probabilities (***prompting***). (d) is the opinion distribution collected from respondents. The lower KL of trainable probe compared to next-token probabilities demonstrates that LLMs know more about opinion than what we observe from their outputs.

et al., 2024; Park et al., 2024a; Suh et al., 2025; Meister et al., 2025; Cao et al., 2025). However, other work—primarily focused on the truthfulness of LLM statements—has shown that LLMs' internal activations sometimes contain more knowledge than is expressed in their outputs (Burns et al., 2022; Li et al., 2023; Azaria & Mitchell, 2023).

**The present work.** Here, we open up the model and ask, what do LLMs *really* know about opinions and where is this knowledge stored? Using two US-representative public opinion datasets, OpinionQA (Santurkar et al., 2023) and SubPOP (Suh et al., 2025), we systematically prompt LLMs (Llama-3.1-8B, Mistral 7B, and Vicuna 7B) with 7,062 prompts corresponding to the product of 22 demographic groups and 321 questions. Instead of only extracting the LLM's response with its next-token probabilities, we also extract its residual stream activations in every internal LLM layer (Figure 1). We then train multinomial probes to predict each group's answer distribution to each question using its residual stream activations, and use Kullback-Leibler (KL) divergence to quantify the distance between the LLM and human response distributions, where a smaller KL divergence indicates greater knowledge of opinions.

First, we show that LLMs know *far* more about opinions than their outputs reveal. Compared to next-token probabilities, the average KL divergence from the best-performing probe (best over layers, chosen based on validation questions) is 59.7% lower on held-out test questions (Figure 2). This result is consistent across all three LLMs we evaluate, linear versus non-linear probes, and different prompting formats. Furthermore, this size of improvement reaches 85% of the improvement achieved by fine-tuning (LoRA) the LLM on the same dataset that the probes were trained on, while training the probe is orders of magnitude cheaper than fine-tuning, with $278\times$ fewer parameters.

Second, we investigate the origin of knowledge discrepancy between probing and next-token probabilities by examining probe performance layer by layer. We find that the knowledge is gained rapidly in the middle layers, with little gain in information after layer 15, but without loss of information as well. Instead, the entire discrepancy is explained by the model unembeddings, which are applied to the final residual stream to compute next-token probabilities. To demonstrate this, we show that fine-tuning *only* the unembeddings achieves a comparable performance to probes, with a 46-62% improvement in KL divergence over next-token probabilities.

Third, we trace where the knowledge of opinions in the residual stream comes from, focusing on attention heads as subspaces where features related to opinions may be encoded. We train sparse autoencoders (SAEs) on attention head activations and find that each demographic group corresponds to clear SAE features that selectively fire only when that group is mentioned in the prompt. Through steerability experiments, we show that manipulating these features per demographic group aligns the model more or less closely with that group, demonstrating that the features are not only correlational but causal. Overall, our study reveals how LLMs encode knowledge of human opinions, contributing to a deeper understanding of their internal mechanisms and enabling broad applications, ranging from more interpretable and human-centered AI to new tools in computational social science.

## 2 TRAINING PROBES TO PREDICT OPINIONS

### 2.1 BACKGROUND

**LLM architecture.** We begin by providing a brief overview of the LLM transformer architecture, using notation similar to Kim et al. (2025). LLMs consist of stacked transformer blocks, where each transformer block modifies the LLM's "residual stream" $\mathbf{x}_\ell$ in layer $\ell$ with the attention heads and multi-layer perceptron (MLP) in that layer (Elhage et al., 2021):

$$\mathbf{x}_\ell = \mathbf{x}_{\ell-1} + \sum_{h=1}^{H} Q_{\ell,h}\mathbf{a}_{\ell,h} + \text{MLP}_\ell\left(\mathbf{x}_{\ell-1} + \sum_{h=1}^{H} Q_{\ell,h}\mathbf{a}_{\ell,h}\right). \tag{1}$$

Here, $\mathbf{a}_{\ell,h}$ refers to the activation of the $h$-th attention head in layer $\ell$:

$$\mathbf{a}_{\ell,h} = \text{ATTN}_{\ell,h}(P_{\ell,h}\mathbf{x}_{\ell-1}). \tag{2}$$

Weight matrices $P_{\ell,h} \in \mathbb{R}^{d_{\ell,h} \times D}$ and $Q_{\ell,h} \in \mathbb{R}^{D \times d_{\ell,h}}$ map between the $D$-dimensional space of the residual stream and the $d_{\ell,h}$-dimensional space of the attention head. Typically, $d_{\ell,h} = d$ is consistent over heads and layers and much smaller than $D$; for example, in the Llama-3.1-8B model, $d$ is 128 and $D$ is 4096.

The initial residual stream $\mathbf{x}_0$ is the embedding of the input tokens, $\mathbf{w}_{<t} = w_1, \cdots, w_{t-1}$. After the forward pass through all the transformer layers, the final residual stream $\mathbf{x}_L$ is mapped back into tokens via the unembedding matrix. The log probability that the next token $w_t$ is $v$ is proportional to the inner product of the unembedding of $v$, $\mathbf{u}_v$, and the final residual stream:

$$P(w_t = v|\mathbf{w}_{<t}) \propto \exp(\mathbf{u}_v \cdot \mathbf{x}_L). \tag{3}$$

**Probing.** Probing is a standard approach for studying the information encoded in a model's internal representations (Alain & Bengio, 2016; Belinkov, 2022). Given a dataset of inputs paired with target labels for a concept of interest, the inputs are passed through the model and the resulting activations are extracted. A lightweight model, or *probe*, is then trained to predict the labels from these activations. Probing enables us to test how much knowledge is encoded in the model and to identify which concepts are captured, based on a careful selection of inputs and labels.

Typically, probes are either trained on the LLM's residual stream or attention head activations. The LLM's residual stream can be seen as the model's memory, reflecting everything the LLM has processed up to that layer, while the attention heads (with much smaller dimensionality) capture specific features that should be attended to and edit the residual stream accordingly (Elhage et al., 2021). In our work, since our primary question is how much the LLM knows about opinions, we start by probing the residual stream, training one probe per layer. Later in Section 4, we turn to the attention heads to decompose this knowledge about opinions into finer grained features, such as ones representing each demographic groups.

A central claim in LLM literature is the linear representation hypothesis, which is the belief that high-level concepts are represented linearly in the LLM's representation space (Park et al., 2024b; Marks & Tegmark, 2023; Gurnee & Tegmark, 2023). A corollary of this hypothesis is that linear probes suffice for predicting concepts from the model's activations and that concepts are not more accurately predicted by more flexible non-linear probes. To test this hypothesis, we evaluate both linear and non-linear probes, specifically a multinomial logistic regression and an MLP.

### 2.2 OUR PROBING FRAMEWORK

In this work, we are interested in the LLM's knowledge of group-specific opinions: given a group $g$ and question $q$, with the associated set of answer options $\mathcal{A}(q)$, what is the group's distribution of answers over $\mathcal{A}(q)$? To test the LLM's knowledge, we construct a prompt $p_{gq}$ that first conditions the model on group $g$, then asks question $q$ (Figure 1a). Prior work using LLMs to predict group-specific opinions has used the same prompting format, but they would extract an answer using the next-token probabilities $P(w_t)$ of the LLM, for the tokens corresponding to the letters of the answer options: $w_t =$ "a", "b", "c", etc (Santurkar et al., 2023; Suh et al., 2025).

We replicate this approach, while also extracting the LLM's residual stream activations from every layer when the LLM is given this prompt (Figure 1b). For a group $g$ and question $q$, let $\mathbf{x}_\ell^{(gq)}$

represent the residual stream activation at layer $\ell$ when the LLM is given the prompt for this group and question, and let $\pi_{gq}$ represent the group's real answer distribution over answer options $\mathcal{A}_q$. We train a probe to learn the function $f_\ell : \mathbb{R}^D \to \mathbb{R}^K$, mapping from $\mathbf{x}_\ell^{(gq)}$ to $\pi_{gq}$ for all groups and all questions with $K$ answer options. Following the linear representation hypothesis, we start with a multinomial logistic regression model, which is a generalized linear model that assumes the log probability of choosing option $k$ has a linear relationship with the input (in this case, the residual stream activation),

$$\hat{\pi}_{gq}[k] = \frac{\exp(\theta_k \cdot \mathbf{x}_\ell^{(gq)})}{\sum_{k' \in [K]} \exp(\theta_{k'} \cdot \mathbf{x}_\ell^{(gq)})}, \tag{4}$$

where we use $[k]$ to index the $k$-th option in $\hat{\pi}$. The learnable parameters in this probe are $\theta_k \in R^D$ per option $k$, resulting in only $D \times K$ parameters, with $D = 4096$ and $K = 2$ or 3. To put the linear representation hypothesis to the test, we also try a non-linear probe, an MLP consisting of a 256-dimensional linear layer followed by a non-linear ReLU activation.

## 2.3 EXPERIMENTS

**Public opinion datasets.** We focus on two datasets: OpinionQA (Santurkar et al., 2023) and Sub-POP (Suh et al., 2025). Both datasets draw from the American Trends Panel (ATP), a nationally representative survey conducted in the US by the Pew Research Center. ATP consists of waves, where each wave is a survey conducted at a certain timepoint covering one or two topics. OpinionQA includes 14 waves and SubPOP includes 53 waves, spanning a wide range of topics such as voter attitudes, news pathways, social media, economic and financial outlook, coronavirus impacts, technology companies and policy issues, online dating, and more (see Table 4 in Suh et al. (2025) for a full list). Both datasets curate the data into the format of (group, question) pairs, providing the group, question text and answer options, and the group's answer distribution for that question. In our experiments, we study 22 demographic groups (e.g., Female) belonging to 8 demographic attributes (e.g., Sex) (see Table 1). While we focus mainly on the OpinionQA and SubPOP datasets in this work to keep the set of groups consistent, we also conduct probing experiments on GlobalOpinionQA (Durmus et al., 2023), which contains country-level responses to cross-national surveys, and show that our main probing results extend to non-US contexts (Appendix A.2).

**Training and evaluating probes.** We train a probe for $K = 2$ (with 603 questions) and $K = 3$ (with 679 questions), for each layer $\ell$ of the LLM ($L = 32$ in all models we evaluate). For each value of $K$, we split all questions with $K$ answer options into train (70%), validation (10%), and test (20%). We train the probe using the input-label pairs ($\mathbf{x}_\ell^{(gq)}$, $\pi_{gq}$) for each of the 22 groups and all training questions. Training is performed with stochastic gradient descent (SGD) with momentum, using a batch size of 32, a learning rate of $6e^{-4}$, and an $\ell_2$ loss. We use the validation questions to tune probe hyperparameters, while we hold out the test questions for evaluation and comparison to next-token probabilities $P(w_t)$. Similar to previous studies, we use Kullback-Leibler divergence (Suh et al., 2025) to quantify the distance between the human distribution $\pi_{gq}$ and the LLM's predicted distribution $\hat{\pi}_{gq}$ (whether using the probe or next-token probabilities):

$$KL(\pi_{gq}, \hat{\pi}_{gq}) = \sum_{k=1}^{K} \pi_{gq}[k] \log \frac{\pi_{gq}[k]}{\hat{\pi}_{gq}[k]}. \tag{5}$$

# 3 WHAT LLMS KNOW ABOUT OPINIONS

## 3.1 INTERNAL KNOWLEDGE VS. OUTPUTS

We first evaluate how much LLMs really know about human opinions. We select the best-performing probes (over all layers) based on the validation set, and evaluate their performance on held-out test questions. These probes substantially outperform next-token probabilities (prompting) at predicting human distributions, revealing that LLMs encode far more about opinions than their outputs suggest. As shown in Figure 2, on average over the 22 demographic groups, we see that KL divergence decreases by 66%, 67%, and 82% for Llama-3.1-8B, Mistral-7B, and Vicuna-7B for questions with

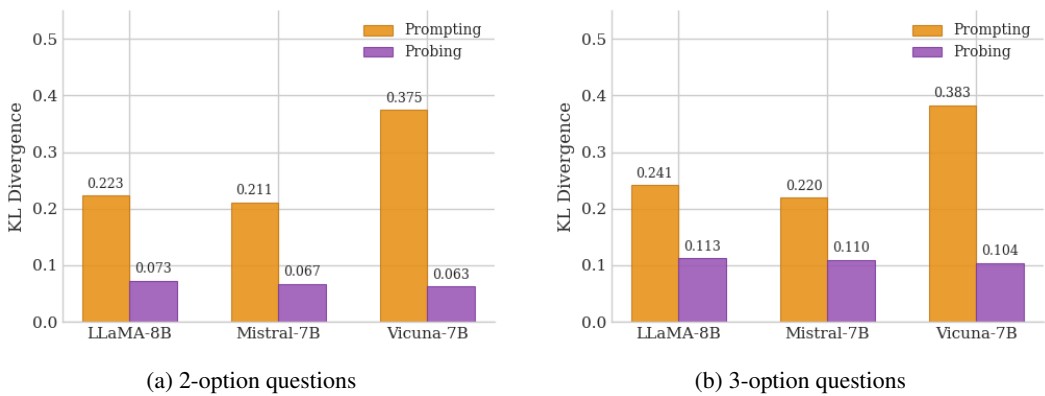

(a) 2-option questions             (b) 3-option questions

Figure 2: Comparing results from prompting (i.e., next-token probabilities) vs. probing.

$K = 2$ options (Figure 2, left), and by 52%, 50%, 71% for the same models, respectively, on questions with $K = 3$ options (Figure 2, right). We provide results per demographic group for all models in the Appendix (Tables 3 and 4).

Second, we test the linear representation hypothesis (Park et al., 2024b) by comparing the performance of the logistic regression vs. MLP probes. Despite the greater expressivity of MLPs, we find that logistic regression probes achieve comparable performance across all layers, models, and 22 groups (Tables 5 and 6), with an average difference in KL of 0.005. While this result is consistent with the linear representation hypothesis, we caution below about what we believe is actually being encoded as the linear concept. Third, we test the sensitivity of our results to the different prompting formats. In addition to our default QA format, which conditions the model on the group in a multiple choice question-answering format (Figure 1a), we also try two other prompting formats used in prior work (Santurkar et al., 2023; Suh et al., 2025): PORTRAY, which instructs the model to answer as if it is a person from that group, and BIO, which describes the group as a first-person biography (see example prompts in Table 2). As shown in Tables 7 and 8, our results are consistent over different prompt formats, where we continue to see large improvements from prompting to probes.

**How can opinions be linear?** While opinions span 22 diverse groups, it is important to distinguish between opinions as an answer distribution vs. as the features involved in building them. By the most predictive layer, the residual stream encodes the answer distribution itself—a simpler object that can be represented linearly—which is precisely what the probe is trained to predict, explaining why logistic regression matches the MLP. Consistent with this, a single shared probe performs nearly identically to per-group probes (Tables 9–10).

If there is a consistent representation of answer distribution that emerges in the residual stream *after* all group and question information is added in, we should see that a single probe for all groups suffices. This is what we see for Llama-3.1-8B, where we find that the probe for all groups performs almost exactly the same as the probe per group, with an average KL difference of 0.003 for 2-option and 0.006 for 3-option questions (Tables 9–10).

**Comparison to fine-tuning.** Prior work showed that fine-tuning LLMs on public opinion survey data achieves large improvements in predicting opinions on unseen questions (Suh et al., 2025; Cao et al., 2025). We are interested in how much of that improvement can be achieved through probing alone, only training the relatively few parameters of the probe and leaving the LLMs' weights untouched. We follow the fine-tuning framework from Suh et al. (2025), where the authors fine-tuned LLMs in the (group, question) format from SubPOP, minimizing the KL divergence between the LLM's next-token probabilities and the ground-truth human distribution $\pi_{gq}$. We use the same LoRA fine-tuning setup as the authors, with the same hyperparameters, and fine-tune Llama-3.1-8B on the same train questions that we used to train our probes.

Consistent with prior work, we find that fine-tuning greatly improves the LLM's ability to predict group-specific opinions on held-out questions (Figure 3). However, probing still achieves over 85% of that improvement, while being $278\times$ more parameter efficient than LoRA. As a practical matter,

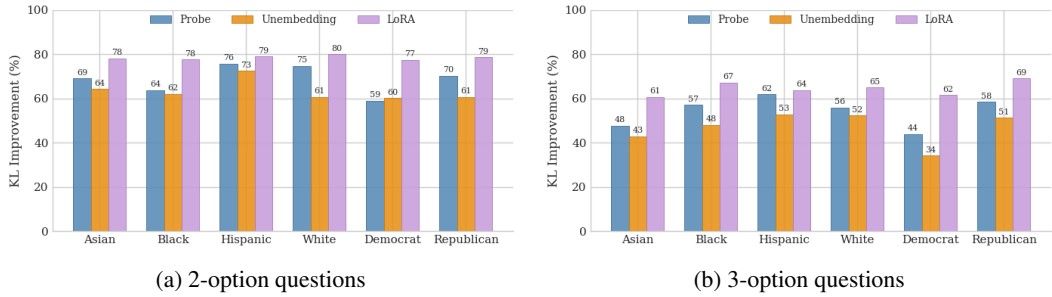

Figure 3: Comparing improvements over prompting from probing, fine-tuning only the unembeddings, and LoRA fine-tuning.

these results reveal that probes could serve as an effective lightweight alternative to fine-tuning. Furthermore, since probes do not teach the LLM new knowledge—they can only use knowledge that is already in the LLM—these results suggest that a majority of the fine-tuning improvement is attributable to the LLM learning how to better retrieve and format knowledge that it already had, as opposed to learning new knowledge, providing evidence for the "Less is More for Alignment" (LIMA) hypothesis (Zhou et al., 2023) in a new domain.

## 3.2 WHERE KNOWLEDGE IS GAINED AND LOST

**Knowledge gains in the first half.** Our second question is where knowledge about opinions is encoded in the LLM. When we examine the layer-by-layer performance of the probes (Figure 4), we see that knowledge is gained in the first half of the model and often with rapid gains in the middle layers (10 to 15), most noticeably for Llama-3.1-8B. This result that opinions are learned in the middle layers contributes to a growing literature exploring where different concepts are located in the model. Other concepts, such as political perspectives (Kim et al., 2025) and sentiment (**?**), have also been shown to emerge in the middle layers, while simpler concepts, such as space and time (Gurnee & Tegmark, 2024), emerge in earlier layers. Furthermore, the rapid gain in knowledge in the residual stream provides clues for which attention heads might be most responsible for adding features about opinions into the residual stream. In Section 4, we look for relevant features in attention heads over all layers, but find that the clearest features corresponding to different demographic groups appear in these middle layers.

**Unembeddings as the source of discrepancy.** After layer 15, probe performance does not improve, but it also does not significantly worsen (Figure 4). Probes trained on the final residual $\mathbf{x}_L$ ($L = 32$) perform as well as the best probes overall, so the gap between probing and prompting cannot be explained by a loss of information in the residual stream. The only component between $\mathbf{x}_L$ and next-token probabilities is the unembedding layer, a linear mapping back to the vocabulary (Eq. 3), meaning that the discrepancy from probe to prompting arises at the unembedding stage. We show this empirically by fine-tuning *only* the unembeddings of Llama-3.1-8B on the same dataset used for training the probes and LoRA fine-tuning. We find that only fine-tuning the unembeddings solves the discrepancy between probing and next-token probabilities. Compared to the next-token probabilities before fine-tuning, the next-token probabilities after fine-tuning the unembeddings have a 61.8% reduction in KL divergence on 2-option questions and a 46.4% reduction in KL divergence on 3-option questions (Tables 11– 12). These improvements are comparable to the improvements we saw from probing (Figure 3), and they also mean that fine-tuning only the final unembeddings achieve 75-78% of the improvement from LoRA fine-tuning, providing another lightweight, targeted alternative to fine-tuning the entire model.

## 4 TRACING KNOWLEDGE BACK TO ATTENTION HEAD FEATURES

So far, we have shown that LLMs contain substantial knowledge about opinions in their residual streams. However, what *features* is the LLM using to arrive at this knowledge? To answer this, we turn to the attention heads, which attend to features and edit the residual stream (Elhage et al., 2021).

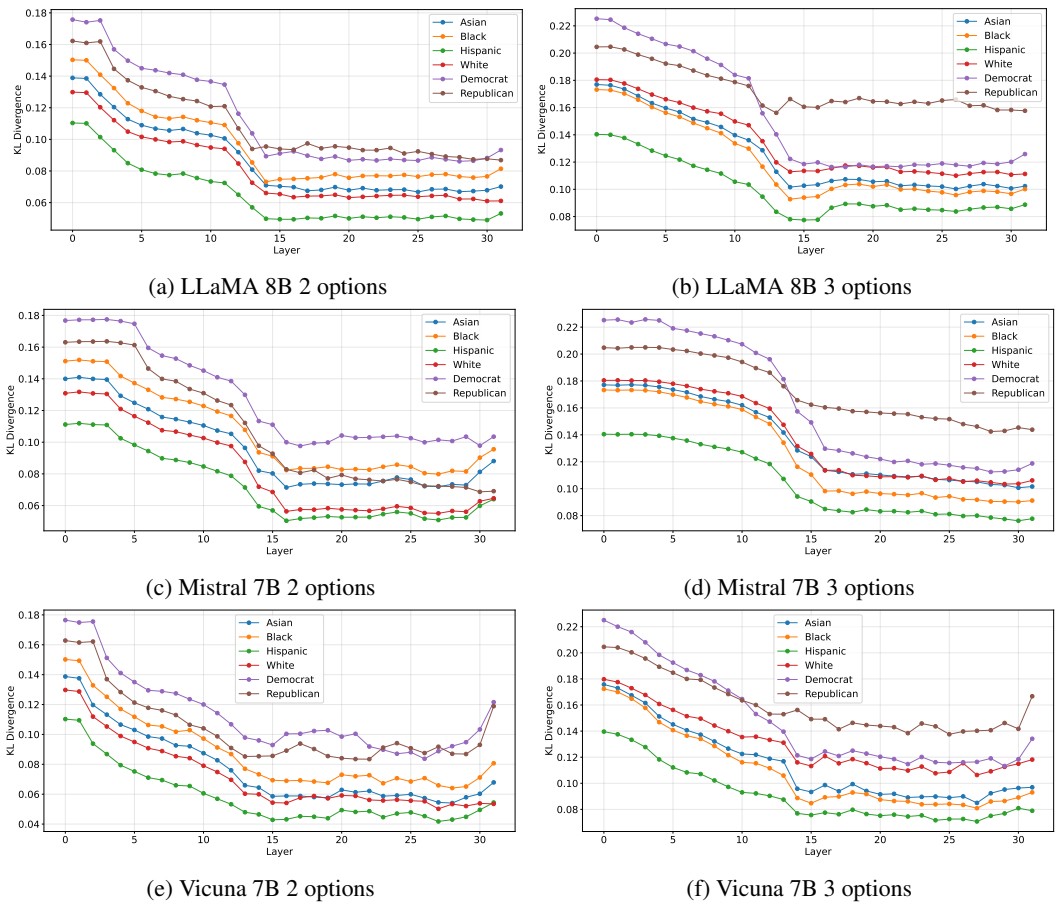

Figure 4: Probe performance per layer for LLaMA (a,b), Mistral (c,d), and Vicuna (e,f) across $K = 2$ (left) and $K = 3$ (right) option questions.

## 4.1 IDENTIFYING FEATURES WITH SAEs

**Background on SAEs.** To identify features in the attention heads, we use sparse autoencoders (SAEs), which are a popular method for decomposing neural network activations into sparse, interpretable features (Cunningham et al., 2024; Kissane et al., 2024; Movva et al., 2025; Gao et al., 2025). We use the $k$-sparse autoencoder, which directly controls the number of active features in the SAE representation using a top-$k$ activation function (Makhzani & Frey, 2013). Given input activations $\mathbf{a}_i \in \mathbb{R}^{D_{\text{in}}}$, the SAE learns a decomposition of $\mathbf{a}_i$ into $M_{\text{SAE}}$ features, where only the top $K_{\text{SAE}}$ features are kept. Using the notation of Movva et al. (2025), the SAE first learns a sparse representation $\mathbf{z}_i \in \mathbb{R}^{M_{\text{SAE}}}$ of $\mathbf{a}_i$:

$$\mathbf{z}_i = \text{ReLU}(\text{TopK}(W_{\text{enc}}(\mathbf{a}_i - \mathbf{b}_{\text{pre}}) + \mathbf{b}_{\text{enc}}), \tag{6}$$

with the encoder $W_{\text{enc}} \in \mathbb{R}^{M_{\text{SAE}} \times D_{\text{in}}}$, bias terms $\mathbf{b}_{\text{pre}}, \mathbf{b}_{\text{enc}} \in \mathbb{R}^{D_{\text{in}}}$, and TopK sets all activations except the top $K_{\text{SAE}}$ to zero. Then, the decoder is applied to $\mathbf{z}_i$ to try to reconstruct $\mathbf{a}_i$: $\hat{\mathbf{a}}_i = W_{\text{dec}}\mathbf{z}_i + \mathbf{b}_{\text{dec}}$, with the decoder $W_{\text{dec}} \in \mathbb{R}^{D_{\text{in}} \times M_{\text{SAE}}}$ and bias term $\mathbf{b}_{\text{dec}} \in \mathbb{R}^{D_{\text{in}}}$. The SAE's loss on a single input is the reconstruction loss: $\mathcal{L}_{\text{SAE}} = ||\mathbf{a}_i - \hat{\mathbf{a}}_i||_2^2$.

**Training our SAEs.** Let $\mathbf{a}_{\ell,h}^{(gq)}$ (as defined in Eq. 2) represent the activation of the $h$-th attention head in layer $\ell$ of the model when it is given the prompt for group $g$ and question $q$. Following Kissane et al. (2024), we train one SAE per model layer on the concatenated activations from all attention heads in that layer, before they are converted to attention outputs through a linear transformation. We focus on the Llama-3.1-8B model for these experiments. We train SAEs on activations extracted at the *demographic token position* — the token corresponding to the demographic group

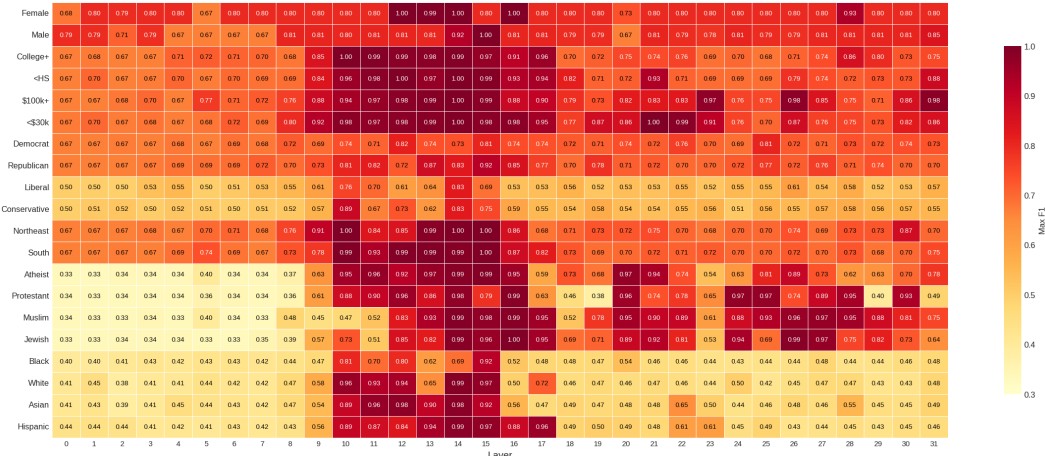

Figure 5: Maximum F1 value achieved by SAE features per demographic group and layer using SAEs with $M_{\text{SAE}} = 2048$ and $K_{\text{SAE}} = 100$.

in the prompt — as opposed to the final token position; we further discuss this in Section A.3. Since the SAE is not tied to the number of answer options, we are able to use the full set of $N = 69,042$ prompts from SubPOP's train dataset, with answer options randomly shuffled to prevent spurious correlations with specific option letters. We split this dataset into 85% for training and 15% for validation. For each model layer $\ell$, we train a separate SAE on activations $a_\ell^{(gq)}$ aggregated across all groups and questions in the training set. To assess model quality, we use standard metrics evaluated on the validation set: reconstruction loss, the percentage of *dead* features (those that never fire), explained variance $(1 - \text{Var}(\mathbf{a} - \hat{\mathbf{a}})/\text{Var}(\mathbf{a}))$, and mean fire count among alive features. The results reported throughout are from SAEs with $M_{\text{SAE}} = 2048$ and $K_{\text{SAE}} = 100$, chosen based on a holistic evaluation of these metrics across SAE sizes and token positions (Table 15).

**Features per demographic group.** After learning the SAE features, we seek to identify the features that correspond most strongly to different demographic groups. For each demographic group, we compute the F1 score of each SAE feature, where prompts that condition on that group serve as the positive class and all other prompts as negatives. In Figure 5, we plot, for each group (y-axis) and model layer (x-axis), the maximum F1 score achieved by SAE features for that group and layer. The pattern is consistent: SAEs trained on middle layers (10–16) recover the largest number of highly group-selective features, with peak F1 scores close to 1.0. SAEs trained on earlier layers also identify features related to demographic groups, but their selectivity is weaker and less consistent. Beyond the middle layers, predictive features become sparse and rarely align with specific groups. Interpreted in terms of the LLM itself, this suggests that demographic group information is first registered early, sharpened into more explicit representations in the middle layers, and largely absent in deeper layers. This layerwise trajectory is consistent with our layer-by-layer probing results on the residual stream (Figure 4), which similarly found that knowledge of opinions emerges rapidly in the middle layers of the LLM.

## 4.2 STEERING TOWARDS DIFFERENT GROUPS

**SAE-based steering methods.** In this final analysis, we test whether the most predictive SAE features are not only correlated with their respective groups, but also causally shift the model's output towards or away from that group. To test this, we design the following procedure, where we intervene on the model by systematically turning SAE features on and off.

1. First, we take a *base* group $g_1$ (e.g., South) and *source* group $g_2$ (e.g., Northeast) from the same attribute (e.g., Region). For a given question $q$, we compute the model's predicted answer distributions $\hat{\pi}_{g_1 q}$ and $\hat{\pi}_{g_2 q}$ when given the prompts $p_{g_1 q}$ and $p_{g_2 q}$, respectively.

2. For a given group, we define a "top" feature as one that satisfies three criteria: (i) F1 $> 0.7$ with respect to that group; (ii) within-attribute selectivity: the feature's firing rate for this group is at

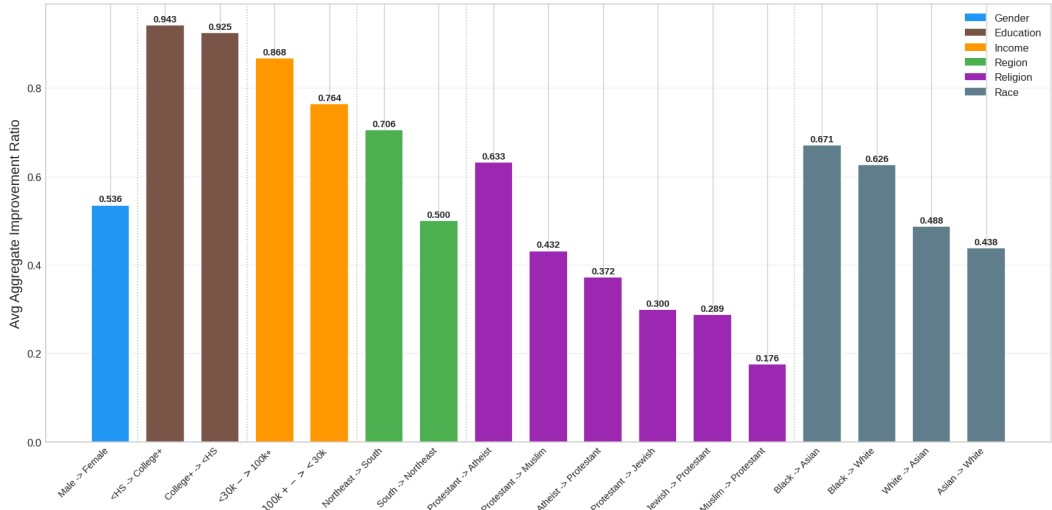

Figure 6: Steering effectiveness across pairs of demographic groups. We report the average improvement ratio, which quantifies the proportion of the original source-to-base distance closed by our intervention.

    least $10\times$ the average firing rate for sibling groups within the same attribute, (iii) cross-attribute selectivity: the feature's firing rate for this group is at least $10\times$ the average firing rate for groups outside of this attribute. A group is included in our steerability analysis only if it has at least 5 top features in total over the layers where we are applying the intervention.

3. We now attempt to shift the model from the base group $g_1$ towards the source group $g_2$. Specifically, for some layer $\ell$, we extract the SAE's representation $\mathbf{z}_1$ and $\mathbf{z}_2$ of the attention head activations when prompted with $p_{g_1 q}$ and $p_{g_2 q}$, respectively. Then, we construct a modified SAE representation $\tilde{\mathbf{z}}^{(\mathbf{1} \rightarrow \mathbf{2})}$ that is a copy of $\mathbf{z}_1$ except the top features for group $g_1$ are turned off (i.e., replaced with 0's) and each top feature $i$ for group $g_2$ is replaced with $\mathbf{z}_2[i] \cdot \delta$, where $\delta > 0$ is some multiplicative factor. This intervention is applied across layers $\ell$ from $\ell_{\text{start}}$ to $\ell_{\text{end}}$, where $\ell_{\text{start}}, \ell_{\text{end}} \in \{0, \ldots, 31\}$.

4. Then, we use the SAE decoder to reconstruct activations in layer $\ell$ from $\tilde{\mathbf{z}}^{(\mathbf{1} \rightarrow \mathbf{2})}$, and run the model forward from $\ell$ with the reconstructed activations and compute the model's new predicted answer distributions, $\tilde{\pi}^{(1 \rightarrow 2)}$. As our evaluation metric, we compare the distance between the source and base distributions, $\mathrm{KL}(\hat{\pi}_{g_2 q}, \hat{\pi}_{g_1 q})$, and between the source and intervened distributions, $\mathrm{KL}(\hat{\pi}_{g_2 q}, \tilde{\pi}^{(1 \rightarrow 2)})$. Our *improvement ratio* is therefore $1 - \mathrm{KL}(\hat{\pi}_{g_2 q}, \tilde{\pi}^{(1 \rightarrow 2)})/\mathrm{KL}(\hat{\pi}_{g_2 q}, \hat{\pi}_{g_1 q})$, i.e., the proportion of the original source-to-base distance closed by the intervention. We sweep over scaling factors $\delta \in \{5, 10, 15, 25\}$ and select the best $\delta$ per demographic group on a fixed 20% tuning split of question keys, stratified across 2- and 3-option questions. All results are reported on the held-out 80% evaluation split.

Notably, our SAE approach enables us to identify features for all 22 demographic groups, in contrast to common steering vector approaches which operate on a single direction between two opposing groups and thus do not naturally extend to settings with more than two groups per attribute (Subramani et al., 2022; Turner et al., 2023; Rimsky et al., 2024).

**Steerability results.** As shown in Figure 6 and Table 16, SAE-based logit steering produces substantial shifts in model behavior. Across the 17 steering directions (layers 9–18), the mean Avg. Improvement is 0.440, with the strongest effects for education and income transitions (e.g., $<$ HS $\rightarrow$ College+: 0.942; $<$ \$30k $\rightarrow$ \$100k+: 0.893), which consistently exceed 0.85. Steering effectiveness varies by attribute type: socioeconomic directions are most amenable, while political ideology directions did not meet the feature selectivity criteria for inclusion. We also observe directional asymmetries within attributes, e.g., Protestant $\rightarrow$ Atheist (0.633) vs. Atheist $\rightarrow$ Protestant (0.372), which correlate with the number of high-F1 features available for each group. A layer-

range ablation further confirms that intervening on layers 9–18 alone achieves within 5% of the full layers 2–30 intervention (0.440 vs. 0.463), consistent with the concentration of demographic-discriminative features in middle layers observed in both the probe accuracy and F1 analyses. Together, these results demonstrate that SAE-identified group features are not merely predictive correlates but actionable, localized levers for targeted behavioral steering.

## 5 RELATED WORK

**Public opinions.**    A growing body of work has examined whether LLMs can predict public opinions, but these studies evaluate only model outputs (Santurkar et al., 2023; Hwang et al., 2023; Chu et al., 2023; Durmus et al., 2023; Moon et al., 2024; Park et al., 2024a; Suh et al., 2025; Meister et al., 2025; Cao et al., 2025). To the best of our knowledge, we are the first to evaluate LLMs' internal knowledge of opinions and to discover the size of the disparity between LLM outputs vs. probes. Our work also contrasts with efforts to improve the model's prediction of group-specific opinions by providing external information, such as few-shot examples (Zhao et al., 2024a) or fine-tuning (Suh et al., 2025; Cao et al., 2025), since our approach seeks to make more out of what is already contained in the model, instead of providing additional information. The closest work to ours probes LLMs on political perspectives, such as the liberal-conservative spectrum (Kim et al., 2025; Hu et al., 2025; Ball et al., 2025), but they do not compare probing to prompting or to fine-tuning. Furthermore, we go far beyond political opinions in our work, studying 22 demographic groups, including non-political variables such as gender, race, and religion that do not fall along a single ideological dimension, along with non-political question topics, such as social identity, education, and cultural preferences. This broader setting requires a more demanding evaluation, since group identities and topics cannot be reduced to a single dimension.

**Interpretability.**    Methodologically, prior work has typically used linear probes to test a scalar value along one or two dimensions (e.g., truth vs. false, liberal vs. conservative) (Burns et al., 2022; Marks & Tegmark, 2024; Gurnee & Tegmark, 2024; Kim et al., 2025). In contrast, our work involves 22 different groups, and we test the model's ability to predict opinion *distributions*, which requires a multinomial probe and distributional metrics, such as KL divergence, to measure model fidelity. Furthermore, common methods for steerability, such as steering vectors, typically rely on a binary contrast (Subramani et al., 2022; Turner et al., 2023; Rimsky et al., 2024). An advantage of our approach is that we can train the SAE once, then steer the model towards any of the 22 groups, by identifying and magnifying the SAE features corresponding to each group. Our layer-wise analysis of where knowledge emerges in the model builds on prior work finding that simpler features emerge in earlier layers (Gurnee & Tegmark, 2024) while more complex features emerge in middle layers (?Kim et al., 2025), and complex processes like multilingualism evolve in the model's functions throughout layers (Zhao et al., 2024b).

## 6 DISCUSSION

Our study demonstrates that LLMs encode far more internal knowledge about opinions than their surface-level outputs suggest. This finding highlights a growing distinction in interpretability research: while probes reveal what models know, outputs reflect only what models are willing to use. Such gaps are crucial for both LLM research and computational social science applications. Our results also suggest practical directions. For instance, domain-specific applications may benefit from fine-tuning only the final layers rather than the entire model, significantly reducing computational and energy costs. Similarly, by identifying where opinion-related knowledge is concentrated in model layers, we may prune irrelevant parameters and design more efficient LLMs.

Despite these advances, our framework has some limitations. First, while we study opinions in US and non-US contexts, we only study English prompts. Second, here we focus on reported opinions from public opinion surveys, following prior work studying opinions (Santurkar et al., 2023; Durmus et al., 2023; Hwang et al., 2023), but public opinion surveys may not capture all underlying human opinions, such as opinions on personal issues or opinions that cannot be expressed in multiple choice answers. Future work could explore how robustly our approach generalizes across cultural and linguistic contexts, to personal opinions, and to open-ended generation.

ACKNOWLEDGEMENTS

The authors thank Ali Shirali, Joseph Suh, and Emma Pierson for helpful comments and the UC Berkeley Center for Human Compatible AI for compute resources. This work is supported in part by the Google Research Scholar Program.

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

Table 1: Attributes in OpinionQA and SubPOP and their corresponding demographic groups.

| | **Race** | **Sex** | **Education** | **Region** | **Religion** | **Income** | **Political Ideology** | **Political Affiliation** |
|---|---|---|---|---|---|---|---|---|
| **Group** | White, Black, Hispanic, Asian | Male, Female | Less Than High School, College Grad | Northeast, South | Protestant, Jewish, Muslim, Hindu, Atheist | Less than $30k, More than $100k | Conservative, Moderate, Liberal | Republican, Democrat |

## A APPENDIX

### A.1 DETAILS FROM PROBING EXPERIMENTS

In this section, we present additional details from our probing experiments, including the 22 groups and 8 attributes included in our experiments (Table 1), examples of different prompting formats (Table 2), comparisons of prompting vs. probing (Tables 3 and 4), comparisons of linear vs. MLP probes (Tables 5 and 6), prompting vs. probing results from the PORTRAY and BIO prompting formats (Tables 7 and 8), comparing one probe per group vs. per attribute vs. shared by all groups (Tables 9 and 10), and comparing probing vs. LoRA fine-tuning vs. only fine-tuning the unembedding (Tables 11 and 12).

Table 2: Examples of three different prompting formats.

| Type | Prompt |
|---|---|
| QA (default) | What is the highest level of schooling or degree that you have completed?
A. Less than high school
B. High school graduate
C. Some college, no degree
D. Associate's degree
E. College graduate/some postgrad
F. Postgraduate
Answer: E

How much, if at all, do you worry about the following happening to you? Being the victim of a terrorist attack
A. Worry a lot
B. Worry a little
C. Do not worry at all
Answer: |
| Bio | Below you will be asked to provide a short description of your current education level and then answer some questions.
Description: The highest level of education I have completed is College graduate/some postgrad

How much, if at all, do you worry about the following happening to you? Being the victim of a terrorist attack
A. Worry a lot
B. Worry a little
C. Do not worry at all
Answer: |
| Portray | Answer the following question as if the highest level of education you have completed is Less than high school

How much, if at all, do you worry about the following happening to you? Being the victim of a terrorist attack
A. Worry a lot
B. Worry a little
C. Do not worry at all
Answer: |

Table 3: Prompt vs. Probing (one row per demographic group). "% Improvement" = (Prompting − Probe)/Prompting × 100. The results are for 2-option questions only.

| Group | LLaMA-8B | | | Mistral-7B | | | Vicuna-7B | | |
|---|---|---|---|---|---|---|---|---|---|
| | Prompt | Probe | % Impro. | Prompt | Probe | % Impro. | Prompt | Probe | % Impro. |
| College graduate/some postgrad | 0.286 | 0.066 | 76.920 | 0.236 | 0.054 | 77.028 | 0.351 | 0.050 | 85.726 |
| Less than high school | 0.190 | 0.049 | 74.553 | 0.173 | 0.045 | 73.905 | 0.367 | 0.047 | 87.282 |
| $100,000 or more | 0.188 | 0.072 | 61.903 | 0.296 | 0.066 | 77.751 | 0.446 | 0.054 | 87.823 |
| Less than $30,000 | 0.181 | 0.046 | 74.409 | 0.171 | 0.048 | 71.807 | 0.432 | 0.047 | 89.224 |
| Conservative | 0.262 | 0.068 | 73.910 | 0.198 | 0.067 | 66.240 | 0.507 | 0.072 | 85.827 |
| Liberal | 0.186 | 0.094 | 49.736 | 0.195 | 0.085 | 56.457 | 0.345 | 0.086 | 75.145 |
| Moderate | 0.188 | 0.053 | 72.006 | 0.208 | 0.053 | 74.749 | 0.357 | 0.051 | 85.607 |
| Democrat | 0.211 | 0.087 | 58.936 | 0.222 | 0.098 | 55.860 | 0.383 | 0.084 | 78.135 |
| Republican | 0.294 | 0.087 | 70.260 | 0.276 | 0.069 | 75.100 | 0.520 | 0.088 | 83.178 |
| Asian | 0.219 | 0.068 | 69.014 | 0.209 | 0.072 | 65.599 | 0.287 | 0.055 | 80.998 |
| Black | 0.211 | 0.077 | 63.672 | 0.220 | 0.080 | 63.649 | 0.383 | 0.066 | 82.808 |
| Hispanic | 0.201 | 0.049 | 75.665 | 0.197 | 0.051 | 74.207 | 0.290 | 0.042 | 85.608 |
| White | 0.241 | 0.061 | 74.637 | 0.180 | 0.055 | 69.392 | 0.384 | 0.050 | 86.934 |
| Northeast | 0.288 | 0.061 | 78.873 | 0.274 | 0.053 | 80.496 | 0.373 | 0.048 | 87.245 |
| South | 0.280 | 0.058 | 79.213 | 0.291 | 0.054 | 81.594 | 0.369 | 0.042 | 88.731 |
| Protestant | 0.234 | 0.071 | 69.615 | 0.161 | 0.055 | 65.667 | 0.372 | 0.061 | 83.525 |
| Jewish | 0.177 | 0.082 | 53.602 | 0.134 | 0.062 | 53.830 | 0.247 | 0.065 | 73.448 |
| Muslim | 0.235 | 0.120 | 48.820 | 0.215 | 0.115 | 46.613 | 0.257 | 0.095 | 63.039 |
| Hindu | 0.196 | 0.089 | 54.372 | 0.184 | 0.084 | 54.552 | 0.282 | 0.077 | 72.550 |
| Atheist | 0.230 | 0.130 | 43.509 | 0.184 | 0.111 | 39.801 | 0.292 | 0.105 | 64.087 |
| Female | 0.189 | 0.058 | 69.117 | 0.199 | 0.046 | 76.888 | 0.563 | 0.055 | 90.285 |
| Male | 0.219 | 0.058 | 73.590 | 0.220 | 0.043 | 80.269 | 0.452 | 0.046 | 89.749 |
| **Average** | 0.223 | 0.073 | 66.651 | 0.211 | 0.067 | 67.339 | 0.375 | 0.063 | 82.134 |

Table 4: Prompt vs. Probing (one row per demographic group). "% Improvement" = (Prompting − Probe)/Prompting × 100. The results are for 3-option questions only.

| Group | LLaMA-8B | | | Mistral-7B | | | Vicuna-7B | | |
|---|---|---|---|---|---|---|---|---|---|
| | Prompt | Probe | % Impro. | Prompt | Probe | % Impro. | Prompt | Probe | % Impro. |
| College graduate/some postgrad | 0.248 | 0.115 | 53.738 | 0.238 | 0.107 | 55.137 | 0.349 | 0.110 | 68.604 |
| Less than high school | 0.216 | 0.068 | 68.433 | 0.160 | 0.060 | 62.776 | 0.313 | 0.061 | 80.698 |
| $100,000 or more | 0.218 | 0.117 | 46.136 | 0.272 | 0.107 | 60.592 | 0.400 | 0.110 | 72.556 |
| Less than $30,000 | 0.240 | 0.072 | 70.095 | 0.177 | 0.069 | 61.038 | 0.371 | 0.066 | 82.338 |
| Conservative | 0.322 | 0.142 | 56.107 | 0.267 | 0.137 | 48.824 | 0.478 | 0.128 | 73.271 |
| Liberal | 0.214 | 0.116 | 45.920 | 0.189 | 0.119 | 37.013 | 0.336 | 0.119 | 64.465 |
| Moderate | 0.204 | 0.087 | 57.396 | 0.170 | 0.088 | 48.180 | 0.382 | 0.081 | 78.880 |
| Democrat | 0.211 | 0.119 | 43.886 | 0.210 | 0.113 | 46.494 | 0.373 | 0.116 | 68.968 |
| Republican | 0.380 | 0.158 | 58.355 | 0.356 | 0.143 | 59.925 | 0.572 | 0.138 | 75.949 |
| Asian | 0.194 | 0.102 | 47.668 | 0.195 | 0.101 | 48.276 | 0.360 | 0.085 | 76.375 |
| Black | 0.216 | 0.093 | 57.133 | 0.210 | 0.090 | 56.960 | 0.346 | 0.081 | 76.597 |
| Hispanic | 0.205 | 0.078 | 61.894 | 0.195 | 0.076 | 60.866 | 0.238 | 0.071 | 70.240 |
| White | 0.256 | 0.113 | 55.898 | 0.213 | 0.104 | 51.326 | 0.435 | 0.106 | 75.519 |
| Northeast | 0.248 | 0.088 | 64.492 | 0.239 | 0.093 | 61.209 | 0.400 | 0.084 | 79.065 |
| South | 0.245 | 0.090 | 63.141 | 0.224 | 0.087 | 61.146 | 0.551 | 0.081 | 85.242 |
| Protestant | 0.269 | 0.108 | 59.896 | 0.207 | 0.107 | 48.290 | 0.384 | 0.104 | 72.924 |
| Jewish | 0.232 | 0.144 | 37.868 | 0.208 | 0.142 | 31.803 | 0.321 | 0.140 | 56.384 |
| Muslim | 0.260 | 0.168 | 35.286 | 0.270 | 0.186 | 31.093 | 0.296 | 0.167 | 43.553 |
| Hindu | 0.249 | 0.168 | 32.374 | 0.250 | 0.172 | 31.082 | 0.293 | 0.141 | 51.884 |
| Atheist | 0.240 | 0.162 | 32.625 | 0.204 | 0.148 | 27.568 | 0.349 | 0.139 | 60.039 |
| Female | 0.206 | 0.094 | 54.516 | 0.186 | 0.090 | 51.531 | 0.475 | 0.080 | 83.121 |
| Male | 0.222 | 0.092 | 58.575 | 0.209 | 0.088 | 57.766 | 0.403 | 0.089 | 77.936 |
| **Average** | 0.241 | 0.113 | 52.792 | 0.220 | 0.110 | 49.950 | 0.383 | 0.104 | 71.573 |

Table 5: Linear vs. MLP probes by demographic group. Difference = Linear − MLP. The results are for 2-option questions only.

| Group | LLaMA-8B | | | Mistral-7B | | | Vicuna-7B | | |
|---|---|---|---|---|---|---|---|---|---|
| | Linear | MLP | Difference | Linear | MLP | Difference | Linear | MLP | Difference |
| College graduate/some postgrad | 0.066 | 0.062 | 0.004 | 0.054 | 0.053 | 0.001 | 0.050 | 0.047 | 0.003 |
| Less than high school | 0.049 | 0.046 | 0.003 | 0.045 | 0.044 | 0.001 | 0.047 | 0.040 | 0.007 |
| $100,000 or more | 0.072 | 0.068 | 0.003 | 0.066 | 0.062 | 0.004 | 0.054 | 0.052 | 0.003 |
| Less than $30,000 | 0.046 | 0.044 | 0.003 | 0.048 | 0.046 | 0.002 | 0.047 | 0.039 | 0.008 |
| Conservative | 0.068 | 0.068 | 0.001 | 0.067 | 0.061 | 0.006 | 0.072 | 0.060 | 0.012 |
| Liberal | 0.094 | 0.087 | 0.006 | 0.085 | 0.080 | 0.005 | 0.086 | 0.069 | 0.017 |
| Moderate | 0.053 | 0.049 | 0.003 | 0.053 | 0.046 | 0.006 | 0.051 | 0.043 | 0.008 |
| Democrat | 0.087 | 0.080 | 0.007 | 0.098 | 0.089 | 0.009 | 0.084 | 0.066 | 0.018 |
| Republican | 0.087 | 0.074 | 0.014 | 0.069 | 0.061 | 0.007 | 0.088 | 0.074 | 0.014 |
| Asian | 0.068 | 0.061 | 0.007 | 0.072 | 0.066 | 0.007 | 0.055 | 0.052 | 0.003 |
| Black | 0.077 | 0.072 | 0.005 | 0.080 | 0.076 | 0.004 | 0.066 | 0.059 | 0.007 |
| Hispanic | 0.049 | 0.044 | 0.005 | 0.051 | 0.045 | 0.006 | 0.042 | 0.041 | 0.001 |
| White | 0.061 | 0.054 | 0.007 | 0.055 | 0.052 | 0.003 | 0.050 | 0.046 | 0.004 |
| Northeast | 0.061 | 0.054 | 0.007 | 0.053 | 0.049 | 0.004 | 0.048 | 0.045 | 0.003 |
| South | 0.058 | 0.051 | 0.007 | 0.054 | 0.048 | 0.006 | 0.042 | 0.041 | 0.000 |
| Protestant | 0.071 | 0.064 | 0.007 | 0.055 | 0.047 | 0.009 | 0.061 | 0.051 | 0.011 |
| Jewish | 0.082 | 0.080 | 0.003 | 0.062 | 0.057 | 0.005 | 0.065 | 0.063 | 0.003 |
| Muslim | 0.120 | 0.115 | 0.005 | 0.115 | 0.108 | 0.007 | 0.095 | 0.097 | −0.002 |
| Hindu | 0.089 | 0.089 | 0.000 | 0.084 | 0.074 | 0.010 | 0.077 | 0.070 | 0.007 |
| Atheist | 0.130 | 0.120 | 0.010 | 0.111 | 0.096 | 0.015 | 0.105 | 0.087 | 0.018 |
| Female | 0.058 | 0.056 | 0.003 | 0.046 | 0.043 | 0.003 | 0.055 | 0.051 | 0.004 |
| Male | 0.058 | 0.056 | 0.002 | 0.043 | 0.041 | 0.002 | 0.046 | 0.046 | 0.001 |
| **Average** | 0.073 | 0.068 | 0.005 | 0.067 | 0.061 | 0.006 | 0.063 | 0.056 | 0.007 |

Table 6: Linear vs. MLP probes by demographic group. Difference = Linear − MLP. The results are for 3-option questions only.

| Group | LLaMA-8B | | | Mistral-7B | | | Vicuna-7B | | |
|---|---|---|---|---|---|---|---|---|---|
| | Linear | MLP | Difference | Linear | MLP | Difference | Linear | MLP | Difference |
| College graduate/some postgrad | 0.115 | 0.111 | 0.004 | 0.107 | 0.104 | 0.003 | 0.110 | 0.103 | 0.007 |
| Less than high school | 0.068 | 0.065 | 0.003 | 0.060 | 0.056 | 0.004 | 0.061 | 0.054 | 0.006 |
| $100,000 or more | 0.117 | 0.113 | 0.004 | 0.107 | 0.102 | 0.005 | 0.110 | 0.106 | 0.004 |
| Less than $30,000 | 0.072 | 0.070 | 0.002 | 0.069 | 0.063 | 0.006 | 0.066 | 0.062 | 0.004 |
| Conservative | 0.142 | 0.139 | 0.002 | 0.137 | 0.130 | 0.007 | 0.128 | 0.112 | 0.016 |
| Liberal | 0.116 | 0.114 | 0.002 | 0.119 | 0.117 | 0.003 | 0.119 | 0.113 | 0.007 |
| Moderate | 0.087 | 0.086 | 0.001 | 0.088 | 0.084 | 0.004 | 0.081 | 0.079 | 0.002 |
| Democrat | 0.119 | 0.109 | 0.010 | 0.113 | 0.108 | 0.005 | 0.116 | 0.104 | 0.012 |
| Republican | 0.158 | 0.137 | 0.021 | 0.143 | 0.136 | 0.007 | 0.138 | 0.126 | 0.012 |
| Asian | 0.102 | 0.097 | 0.005 | 0.101 | 0.095 | 0.006 | 0.085 | 0.081 | 0.004 |
| Black | 0.093 | 0.092 | 0.000 | 0.090 | 0.087 | 0.003 | 0.081 | 0.079 | 0.002 |
| Hispanic | 0.078 | 0.081 | −0.003 | 0.076 | 0.072 | 0.004 | 0.071 | 0.068 | 0.003 |
| White | 0.113 | 0.104 | 0.008 | 0.104 | 0.099 | 0.004 | 0.106 | 0.097 | 0.009 |
| Northeast | 0.088 | 0.087 | 0.001 | 0.093 | 0.093 | 0.000 | 0.084 | 0.079 | 0.004 |
| South | 0.090 | 0.089 | 0.002 | 0.087 | 0.087 | 0.000 | 0.081 | 0.076 | 0.005 |
| Protestant | 0.108 | 0.108 | 0.000 | 0.107 | 0.104 | 0.003 | 0.104 | 0.090 | 0.014 |
| Jewish | 0.144 | 0.141 | 0.004 | 0.142 | 0.141 | 0.001 | 0.140 | 0.134 | 0.006 |
| Muslim | 0.168 | 0.166 | 0.003 | 0.186 | 0.186 | 0.000 | 0.167 | 0.169 | −0.002 |
| Hindu | 0.168 | 0.166 | 0.002 | 0.172 | 0.170 | 0.003 | 0.141 | 0.148 | −0.007 |
| Atheist | 0.162 | 0.150 | 0.012 | 0.148 | 0.143 | 0.005 | 0.139 | 0.132 | 0.007 |
| Female | 0.094 | 0.091 | 0.003 | 0.090 | 0.090 | 0.000 | 0.080 | 0.079 | 0.001 |
| Male | 0.092 | 0.088 | 0.004 | 0.088 | 0.088 | 0.000 | 0.089 | 0.082 | 0.007 |
| **Average** | 0.113 | 0.109 | 0.004 | 0.110 | 0.107 | 0.003 | 0.104 | 0.099 | 0.006 |

Table 7: Prompting vs. probing results with two alternate prompting formats, PORTRAY and BIO, as described in Table 2. "% Improvement" = (Prompting − Probe)/Prompting × 100. The results are for 2-option questions only, using the Llama-3.1-8B model.

| Group | QA | | | BIO | | | PORTRAY | | |
|---|---|---|---|---|---|---|---|---|---|
| | Prompt | Probe | % Impro. | Prompt | Probe | % Impro. | Prompt | Probe | % Impro. |
| College graduate/some postgrad | 0.286 | 0.066 | 76.920 | 0.142 | 0.061 | 57.184 | 0.174 | 0.058 | 66.921 |
| Less than high school | 0.190 | 0.049 | 74.553 | 0.104 | 0.045 | 56.331 | 0.121 | 0.041 | 65.878 |
| $100,000 or more | 0.188 | 0.072 | 61.903 | 0.151 | 0.063 | 57.882 | 0.189 | 0.067 | 64.762 |
| Less than $30,000 | 0.181 | 0.046 | 74.409 | 0.114 | 0.050 | 56.003 | 0.169 | 0.051 | 70.032 |
| Conservative | 0.262 | 0.068 | 73.910 | 0.167 | 0.072 | 56.927 | 0.239 | 0.069 | 71.122 |
| Liberal | 0.186 | 0.094 | 49.736 | 0.153 | 0.074 | 51.670 | 0.224 | 0.081 | 63.856 |
| Moderate | 0.188 | 0.053 | 72.006 | 0.157 | 0.054 | 65.471 | 0.217 | 0.057 | 73.739 |
| Democrat | 0.211 | 0.087 | 58.936 | 0.146 | 0.077 | 47.265 | 0.167 | 0.074 | 55.854 |
| Republican | 0.294 | 0.087 | 70.260 | 0.180 | 0.084 | 53.186 | 0.225 | 0.091 | 59.655 |
| Asian | 0.219 | 0.068 | 69.014 | 0.142 | 0.063 | 55.566 | 0.169 | 0.060 | 64.484 |
| Black | 0.211 | 0.077 | 63.672 | 0.153 | 0.071 | 53.616 | 0.172 | 0.071 | 58.724 |
| Hispanic | 0.201 | 0.049 | 75.665 | 0.121 | 0.047 | 60.998 | 0.126 | 0.043 | 65.792 |
| White | 0.241 | 0.061 | 74.637 | 0.133 | 0.058 | 56.507 | 0.171 | 0.056 | 67.246 |
| Northeast | 0.288 | 0.061 | 78.873 | 0.169 | 0.053 | 68.906 | 0.157 | 0.051 | 67.526 |
| South | 0.280 | 0.058 | 79.213 | 0.156 | 0.047 | 69.903 | 0.169 | 0.046 | 73.029 |
| Protestant | 0.234 | 0.071 | 69.615 | 0.119 | 0.064 | 46.634 | 0.175 | 0.063 | 63.847 |
| Jewish | 0.177 | 0.082 | 53.602 | 0.114 | 0.068 | 40.717 | 0.165 | 0.072 | 56.377 |
| Muslim | 0.235 | 0.120 | 48.820 | 0.162 | 0.106 | 34.493 | 0.205 | 0.103 | 49.798 |
| Hindu | 0.196 | 0.089 | 54.372 | 0.141 | 0.094 | 33.576 | 0.185 | 0.088 | 52.474 |
| Atheist | 0.230 | 0.130 | 43.509 | 0.154 | 0.110 | 28.293 | 0.199 | 0.099 | 50.464 |
| Female | 0.189 | 0.058 | 69.117 | 0.139 | 0.056 | 60.056 | 0.183 | 0.048 | 73.642 |
| Male | 0.219 | 0.058 | 73.590 | 0.140 | 0.054 | 61.685 | 0.185 | 0.046 | 74.938 |
| **Average** | 0.223 | 0.073 | 66.651 | 0.143 | 0.067 | 53.312 | 0.181 | 0.065 | 64.098 |

Table 8: Prompting vs. probing results with two alternate prompting formats, PORTRAY and BIO, as described in Table 2. "% Improvement" = (Prompting − Probe)/Prompting × 100. The results are for 3-option questions only, using the Llama-3.1-8B model.

| Group | QA | | | BIO | | | PORTRAY | | |
|---|---|---|---|---|---|---|---|---|---|
| | Prompt | Probe | % Impro. | Prompt | Probe | % Impro. | Prompt | Probe | % Impro. |
| College graduate/some postgrad | 0.248 | 0.115 | 53.738 | 0.189 | 0.101 | 46.383 | 0.207 | 0.118 | 42.844 |
| Less than high school | 0.216 | 0.068 | 68.433 | 0.153 | 0.058 | 62.024 | 0.167 | 0.072 | 57.129 |
| $100,000 or more | 0.218 | 0.117 | 46.136 | 0.223 | 0.111 | 50.093 | 0.230 | 0.126 | 45.374 |
| Less than $30,000 | 0.240 | 0.072 | 70.095 | 0.152 | 0.069 | 54.942 | 0.195 | 0.083 | 57.377 |
| Conservative | 0.322 | 0.142 | 56.107 | 0.233 | 0.128 | 44.981 | 0.300 | 0.131 | 56.161 |
| Liberal | 0.214 | 0.116 | 45.920 | 0.177 | 0.113 | 36.416 | 0.247 | 0.137 | 44.357 |
| Moderate | 0.204 | 0.087 | 57.396 | 0.168 | 0.085 | 49.513 | 0.211 | 0.093 | 56.207 |
| Democrat | 0.211 | 0.119 | 43.886 | 0.162 | 0.102 | 36.710 | 0.208 | 0.123 | 40.945 |
| Republican | 0.380 | 0.158 | 58.355 | 0.241 | 0.144 | 40.411 | 0.298 | 0.159 | 46.503 |
| Asian | 0.194 | 0.102 | 47.668 | 0.167 | 0.100 | 40.290 | 0.193 | 0.111 | 42.134 |
| Black | 0.216 | 0.093 | 57.133 | 0.163 | 0.086 | 47.138 | 0.209 | 0.099 | 52.619 |
| Hispanic | 0.205 | 0.078 | 61.894 | 0.144 | 0.076 | 46.924 | 0.174 | 0.086 | 50.434 |
| White | 0.256 | 0.113 | 55.898 | 0.186 | 0.104 | 43.989 | 0.242 | 0.117 | 51.564 |
| Northeast | 0.248 | 0.088 | 64.492 | 0.171 | 0.084 | 50.907 | 0.185 | 0.091 | 50.961 |
| South | 0.245 | 0.090 | 63.141 | 0.168 | 0.080 | 52.331 | 0.188 | 0.086 | 54.562 |
| Protestant | 0.269 | 0.108 | 59.896 | 0.179 | 0.112 | 37.506 | 0.224 | 0.117 | 47.878 |
| Jewish | 0.232 | 0.144 | 37.868 | 0.209 | 0.144 | 31.084 | 0.221 | 0.159 | 28.200 |
| Muslim | 0.260 | 0.168 | 35.286 | 0.258 | 0.182 | 29.446 | 0.255 | 0.178 | 30.208 |
| Hindu | 0.249 | 0.168 | 32.374 | 0.234 | 0.165 | 29.708 | 0.226 | 0.176 | 22.104 |
| Atheist | 0.240 | 0.162 | 32.625 | 0.248 | 0.155 | 37.372 | 0.235 | 0.170 | 27.467 |
| Female | 0.206 | 0.094 | 54.516 | 0.170 | 0.084 | 50.533 | 0.210 | 0.099 | 52.562 |
| Male | 0.222 | 0.092 | 58.575 | 0.169 | 0.087 | 48.852 | 0.192 | 0.097 | 49.692 |
| **Average** | 0.241 | 0.113 | 52.792 | 0.189 | 0.108 | 43.980 | 0.219 | 0.119 | 45.786 |

Table 9: Group-level vs. Single/All Attribute Probes (2-option questions, linear probe), using the Llama-3.1-8B model

| Group | Group Level | Single Attribute | All Attributes | Single - Group | All - Group |
|---|---|---|---|---|---|
| College graduate/some postgrad | 0.067 | 0.066 | 0.064 | −0.001 | −0.003 |
| Less than high school | 0.049 | 0.049 | 0.049 | 0.000 | 0.001 |
| $100,000 or more | 0.071 | 0.072 | 0.074 | 0.000 | 0.003 |
| Less than $30,000 | 0.047 | 0.046 | 0.049 | −0.001 | 0.002 |
| Conservative | 0.065 | 0.068 | 0.080 | 0.004 | 0.016 |
| Liberal | 0.090 | 0.094 | 0.095 | 0.003 | 0.005 |
| Moderate | 0.052 | 0.053 | 0.056 | 0.001 | 0.004 |
| Democrat | 0.081 | 0.087 | 0.089 | 0.006 | 0.009 |
| Republican | 0.077 | 0.087 | 0.098 | 0.011 | 0.022 |
| Asian | 0.067 | 0.068 | 0.065 | 0.001 | −0.001 |
| Black | 0.072 | 0.077 | 0.076 | 0.005 | 0.004 |
| Hispanic | 0.050 | 0.049 | 0.046 | −0.001 | −0.004 |
| White | 0.059 | 0.061 | 0.055 | 0.003 | −0.004 |
| Northeast | 0.062 | 0.061 | 0.058 | −0.001 | −0.004 |
| South | 0.057 | 0.058 | 0.054 | 0.001 | −0.003 |
| Protestant | 0.058 | 0.071 | 0.059 | 0.014 | 0.001 |
| Jewish | 0.083 | 0.082 | 0.080 | −0.001 | −0.003 |
| Muslim | 0.123 | 0.120 | 0.122 | −0.003 | −0.001 |
| Hindu | 0.091 | 0.089 | 0.095 | −0.001 | 0.005 |
| Atheist | 0.121 | 0.130 | 0.135 | 0.008 | 0.014 |
| Female | 0.057 | 0.058 | 0.059 | 0.001 | 0.002 |
| Male | 0.058 | 0.058 | 0.059 | −0.001 | 0.001 |
| **Average** | 0.071 | 0.073 | 0.074 | 0.002 | 0.003 |

Table 10: Group-level vs. Single/All Attribute Probes (3-option questions, linear probe), using the Llama-3.1-8B model.

| Group | Group Level | Single Attribute | All Attributes | Single - Group | All - Group |
|---|---|---|---|---|---|
| College graduate/some postgrad | 0.114 | 0.115 | 0.118 | 0.001 | 0.005 |
| Less than high school | 0.062 | 0.068 | 0.076 | 0.006 | 0.014 |
| $100,000 or more | 0.116 | 0.117 | 0.118 | 0.001 | 0.001 |
| Less than $30,000 | 0.070 | 0.072 | 0.072 | 0.002 | 0.002 |
| Conservative | 0.129 | 0.142 | 0.145 | 0.013 | 0.016 |
| Liberal | 0.111 | 0.116 | 0.114 | 0.005 | 0.003 |
| Moderate | 0.087 | 0.087 | 0.088 | −0.001 | 0.000 |
| Democrat | 0.103 | 0.119 | 0.113 | 0.016 | 0.010 |
| Republican | 0.136 | 0.158 | 0.164 | 0.022 | 0.028 |
| Asian | 0.100 | 0.102 | 0.103 | 0.002 | 0.003 |
| Black | 0.088 | 0.093 | 0.102 | 0.005 | 0.014 |
| Hispanic | 0.078 | 0.078 | 0.088 | 0.000 | 0.009 |
| White | 0.105 | 0.113 | 0.112 | 0.008 | 0.007 |
| Northeast | 0.090 | 0.088 | 0.089 | −0.002 | −0.001 |
| South | 0.088 | 0.090 | 0.093 | 0.002 | 0.005 |
| Protestant | 0.103 | 0.108 | 0.107 | 0.005 | 0.004 |
| Jewish | 0.149 | 0.144 | 0.141 | −0.005 | −0.009 |
| Muslim | 0.166 | 0.168 | 0.170 | 0.002 | 0.004 |
| Hindu | 0.173 | 0.168 | 0.173 | −0.005 | 0.000 |
| Atheist | 0.150 | 0.162 | 0.154 | 0.011 | 0.004 |
| Female | 0.095 | 0.094 | 0.099 | −0.002 | 0.004 |
| Male | 0.092 | 0.092 | 0.100 | 0.000 | 0.008 |
| **Average** | 0.109 | 0.113 | 0.115 | 0.004 | 0.006 |

Table 11: Unembedding fine-tuning and LoRA fine-tuning for 2-option questions (LLaMA), using the Llama-3.1-8B model. Improvements are computed relative to prompting: Imp. = (Prompt KL − Method KL)/Prompt KL.

| Group | Prompt KL | KL (Unemb) | Imp. (Unemb) | KL (LoRA) | Imp. (LoRA) | Imp(Unemb)/Imp(LoRA) |
|---|---|---|---|---|---|---|
| College grad/some postgrad | 0.280 | 0.087 | 0.689 | 0.049 | 0.825 | 0.835 |
| Less than high school | 0.189 | 0.076 | 0.598 | 0.044 | 0.767 | 0.779 |
| $100,000 or more | 0.192 | 0.068 | 0.647 | 0.041 | 0.787 | 0.822 |
| Less than $30,000 | 0.191 | 0.063 | 0.670 | 0.040 | 0.791 | 0.848 |
| Male | 0.208 | 0.073 | 0.648 | 0.044 | 0.788 | 0.823 |
| Female | 0.178 | 0.084 | 0.527 | 0.038 | 0.786 | 0.671 |
| White | 0.223 | 0.088 | 0.606 | 0.045 | 0.798 | 0.759 |
| Black | 0.200 | 0.076 | 0.621 | 0.045 | 0.775 | 0.800 |
| Hispanic | 0.189 | 0.052 | 0.725 | 0.040 | 0.789 | 0.920 |
| Asian | 0.196 | 0.070 | 0.643 | 0.043 | 0.781 | 0.824 |
| Protestant | 0.215 | 0.069 | 0.679 | 0.039 | 0.819 | 0.829 |
| Jewish | 0.186 | 0.095 | 0.489 | 0.047 | 0.747 | 0.654 |
| Muslim | 0.214 | 0.087 | 0.593 | 0.049 | 0.771 | 0.770 |
| Hindu | 0.204 | 0.125 | 0.386 | 0.058 | 0.715 | 0.540 |
| Atheist | 0.233 | 0.126 | 0.460 | 0.056 | 0.760 | 0.605 |
| Liberal | 0.204 | 0.081 | 0.604 | 0.047 | 0.770 | 0.784 |
| Conservative | 0.231 | 0.076 | 0.671 | 0.048 | 0.792 | 0.847 |
| Moderate | 0.197 | 0.074 | 0.624 | 0.045 | 0.771 | 0.809 |
| Democrat | 0.204 | 0.081 | 0.602 | 0.046 | 0.774 | 0.778 |
| Republican | 0.239 | 0.094 | 0.606 | 0.051 | 0.786 | 0.771 |
| Northeast | 0.275 | 0.086 | 0.687 | 0.050 | 0.818 | 0.840 |
| South | 0.249 | 0.063 | 0.747 | 0.037 | 0.851 | 0.877 |
| **Average** | 0.213 | 0.082 | 0.618 | 0.046 | 0.787 | 0.786 |

Table 12: Unembedding finetuning and LoRA finetuning for 3-option questions (LLaMA), using the Llama-3.1-8B model. Improvements are computed relative to prompting: Imp. = (Prompt KL − Method KL)/Prompt KL.

| Group | Prompt KL | KL (Unemb) | Imp. (Unemb) | KL (LoRA) | Imp. (LoRA) | Imp(Unemb.)/Imp(LoRA) |
|---|---|---|---|---|---|---|
| College grad/some postgrad | 0.250 | 0.114 | 0.544 | 0.095 | 0.620 | 0.877 |
| Less than high school | 0.198 | 0.083 | 0.580 | 0.062 | 0.686 | 0.845 |
| $100,000 or more | 0.209 | 0.134 | 0.360 | 0.098 | 0.532 | 0.677 |
| Less than $30,000 | 0.227 | 0.088 | 0.612 | 0.056 | 0.753 | 0.813 |
| Male | 0.196 | 0.105 | 0.464 | 0.075 | 0.617 | 0.752 |
| Female | 0.200 | 0.107 | 0.464 | 0.078 | 0.609 | 0.762 |
| White | 0.239 | 0.114 | 0.523 | 0.084 | 0.648 | 0.806 |
| Black | 0.213 | 0.111 | 0.480 | 0.070 | 0.672 | 0.714 |
| Hispanic | 0.201 | 0.095 | 0.527 | 0.073 | 0.637 | 0.828 |
| Asian | 0.208 | 0.119 | 0.428 | 0.082 | 0.606 | 0.707 |
| Protestant | 0.237 | 0.105 | 0.557 | 0.084 | 0.646 | 0.863 |
| Jewish | 0.206 | 0.146 | 0.292 | 0.111 | 0.462 | 0.633 |
| Muslim | 0.279 | 0.184 | 0.340 | 0.149 | 0.465 | 0.730 |
| Hindu | 0.278 | 0.198 | 0.287 | 0.152 | 0.453 | 0.635 |
| Atheist | 0.236 | 0.173 | 0.266 | 0.135 | 0.427 | 0.622 |
| Liberal | 0.241 | 0.138 | 0.427 | 0.094 | 0.610 | 0.700 |
| Conservative | 0.299 | 0.143 | 0.522 | 0.099 | 0.669 | 0.780 |
| Moderate | 0.219 | 0.111 | 0.494 | 0.073 | 0.667 | 0.740 |
| Democrat | 0.208 | 0.137 | 0.342 | 0.080 | 0.616 | 0.556 |
| Republican | 0.326 | 0.159 | 0.513 | 0.101 | 0.690 | 0.742 |
| Northeast | 0.250 | 0.105 | 0.579 | 0.075 | 0.699 | 0.828 |
| South | 0.236 | 0.095 | 0.597 | 0.068 | 0.711 | 0.839 |
| **Average** | 0.234 | 0.126 | 0.464 | 0.091 | 0.613 | 0.756 |

## A.2 GENERALIZATION TO NON-US CONTEXTS

To test whether our results generalize to non-US contexts, we apply the same probing framework to GlobalOpinionQA (Durmus et al., 2023). GlobalOpinionQA contains questions from the Pew Global Attitudes Survey and the World Values Survey (WVS), with responses from countries around the globe. The dataset is curated as (country, question) pairs, where each pair gives a question and the corresponding answer distribution for a country. Instead of predicting the opinions of demographic groups within the US, we predict the opinions of different countries on cross-national surveys. We keep the top 5 countries in GlobalOpinionQA besides the US and fit a probe to predict their opinion distributions. We find similar results in this setting: probing achieves a significantly lower KL than next-token probabilities (26-39% reduction, as shown in Tables 13 and 14), and we observe a similar layer-wise trend, with knowledge emerging rapidly in the middle layers (Figure 7).

Table 13: Prompt vs. Probing by country (2-option questions). "% Improvement" = (Prompt − Probe)/Prompt × 100.

| Country | Prompt KL | Probe KL | Improvement (%) |
|---------|-----------|----------|-----------------|
| Germany | 0.135 | 0.101 | 25.440 |
| France | 0.106 | 0.077 | 27.950 |
| Britain | 0.130 | 0.070 | 45.940 |
| Spain | 0.156 | 0.105 | 32.860 |
| Turkey | 0.225 | 0.108 | 52.110 |
| **Average** | 0.150 | 0.092 | 38.860 |

Table 14: Prompt vs. Probing by country (3-option questions). "% Improvement" = (Prompt − Probe)/Prompt × 100.

| Country | Prompt KL | Probe KL | Improvement (%) |
|---------|-----------|----------|-----------------|
| Germany | 0.196 | 0.142 | 27.950 |
| France | 0.196 | 0.165 | 15.880 |
| Britain | 0.183 | 0.142 | 22.200 |
| Spain | 0.134 | 0.125 | 6.650 |
| Turkey | 0.254 | 0.136 | 46.380 |
| **Average** | 0.193 | 0.142 | 26.290 |

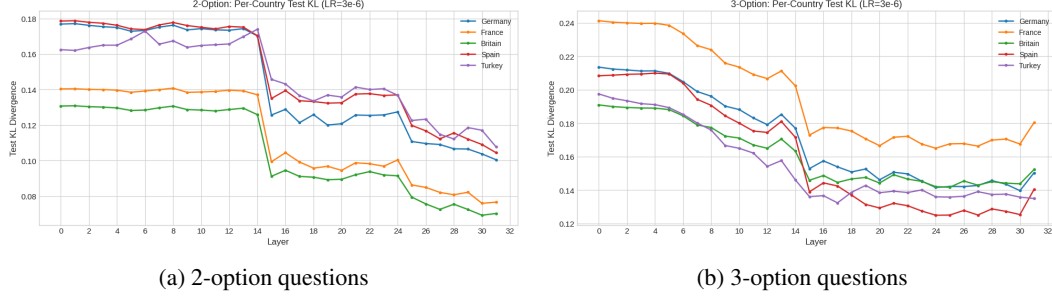

(a) 2-option questions          (b) 3-option questions

Figure 7: Layer-by-layer probe performance on GlobalOpinionQA evaluated on the held-out set.

A.3   SPARSE AUTOENCODER AND STEERABILITY DETAILS

In this section, we provide implementation details for our sparse autoencoder (SAE) analysis and motivate our final configuration choice.

Each attention head in Llama-3.1-8B has dimensionality $d = 128$, so the concatenated output of all 32 heads per layer has dimensionality $D_{\text{in}} = 32d = 4096$. We train SAEs on these concatenated head activations per layer. We compare six configurations spanning three dictionary widths and two token positions:

$$(M_{\text{SAE}}, K_{\text{SAE}}) \in \{(512, 32), (2048, 100), (4096, 100)\}$$

evaluated at either the demographic-label token (demo) or the final token (last). Larger widths are paired with higher top-$k$ values to maintain comparable reconstruction quality.

**Configuration comparison.**   We evaluate six SAE configurations (three dictionary widths $\times$ two token positions) by computing, for each of the 22 demographic groups, the maximum F1 score achieved by any feature at each layer, reported in Figures 8- 12.

*Token position.* SAEs trained on the demographic-label token substantially outperform those trained on the final token. Demo-token SAEs achieve mean max-F1 scores of 0.622–0.693 across widths, with 448–2,494 features exceeding F1 $\geq 0.7$. In contrast, last-token SAEs achieve mean max-F1 of only 0.525–0.553 and just 27–70 features above the same threshold—an order-of-magnitude reduction—despite comparable reconstruction quality (explained variance 80.6–88.8% vs. 98.2–99.0%). This indicates that demographic identity is primarily encoded at the token position where the group label appears, rather than being propagated to the final position.

*Dictionary width.* Among demo-token SAEs, $M_{\text{SAE}} = 2048$ and $K_{\text{SAE}} = 100$ achieves the highest mean max-F1 (0.693) and 1,934 features above F1 $\geq 0.7$, compared to 0.622/448 for $M_{\text{SAE}} = 512$, $K_{\text{SAE}} = 32$ and 0.669/2,494 for $M_{\text{SAE}} = 4096$, $K_{\text{SAE}} = 100$. Although $M_{\text{SAE}} = 4096$, $K_{\text{SAE}} = 100$ recovers more total features, its mean F1 is lower and it exhibits a substantially higher dead-feature rate (61.3% vs. 43.0%), indicating inefficient dictionary utilization. The smallest model ($M_{\text{SAE}} = 512$, $K_{\text{SAE}} = 32$) has the lowest dead rate (38.2%) but insufficient capacity to disentangle group-specific structure, yielding the lowest mean F1.

**Selected configuration.**   Based on this analysis, we use the demo-token SAE with $M_{\text{SAE}} = 2048$ and $K_{\text{SAE}} = 100$ in all steerability experiments. This configuration provides the best balance of feature discriminability (highest mean max-F1), broad feature coverage across groups and layers, moderate dead-feature rate, and strong reconstruction fidelity (98.95% explained variance; MSE = 0.0105). We provide further details about the SAEs metrics and the feature qualities in Table 15 and Figures **??**- 12 .

**Layer range ablation.**   We compare two intervention ranges: a focused range spanning layers 9–18 (Table 16) and a broader range spanning layers 2–30 (Table 17). Both tables report results across the same 17 steering directions based on experiment passing the feature quality criteria.

The focused 9–18 intervention achieves a mean aggregate improvement of 0.537 across directions, while the broader 2–30 intervention achieves 0.561, a difference of 0.024 (approximately 4%). Despite intervening on nearly three times as many layers (28 vs. 10), the broader range yields only a modest increase in average steering performance.

This close correspondence indicates that the majority of steering signal is concentrated within the middle layers. We therefore adopt the focused 9–18 intervention range in our main experiments, as it substantially reduces computational cost while maintaining comparable overall performance.

Table 15: SAE metrics summary (mean across layers 0–31).

| Token Type | Width | Best Val MSE | Explained Var | Dead % | Mean Fire Count (alive) | Mean Recon Strength |
|---|---|---|---|---|---|---|
| demo | w512_k32 | 0.018 | 98.22% | 38.2% | 1100.900 | 6.569 |
| demo | w2048_k100 | 0.011 | 98.95% | 43.0% | 935.100 | 2.924 |
| demo | w4096_k100 | 0.010 | 99.02% | 61.3% | 692.000 | 2.398 |
| last | w512_k32 | 0.194 | 80.61% | 26.9% | 943.100 | 7.025 |
| last | w2048_k100 | 0.116 | 88.36% | 26.3% | 724.100 | 3.284 |
| last | w4096_k100 | 0.112 | 88.83% | 44.4% | 483.500 | 3.048 |

Table 16: Steerability experiments across layers 9-18. $\delta$ denotes the steering strength. "Num Target Features" is the number of features used for the target group. "Avg. Improvement" averages 2-option and 3-option improvements.

| Experiment | Attribute | $\delta$ | Num Target Features | Avg. Improvement |
|---|---|---|---|---|
| Less than high school → College graduate/some postgrad | Education | 10.000 | 45.000 | 0.943 |
| College graduate/some postgrad → Less than high school | Education | 10.000 | 46.000 | 0.925 |
| Less than $30,000 → $100,000 or more | Income | 10.000 | 49.000 | 0.868 |
| $100,000 or more → Less than $30,000 | Income | 10.000 | 48.000 | 0.764 |
| Northeast → South | Region | 15.000 | 19.000 | 0.706 |
| Black → Asian | Race | 15.000 | 17.000 | 0.671 |
| Protestant → Atheist | Religion | 10.000 | 34.000 | 0.633 |
| Black → White | Race | 15.000 | 16.000 | 0.626 |
| Male → Female | Gender | 25.000 | 6.000 | 0.536 |
| South → Northeast | Region | 10.000 | 12.000 | 0.500 |
| White → Asian | Race | 15.000 | 17.000 | 0.488 |
| Asian → White | Race | 10.000 | 16.000 | 0.438 |
| Protestant → Muslim | Religion | 15.000 | 23.000 | 0.432 |
| Atheist → Protestant | Religion | 10.000 | 23.000 | 0.372 |
| Protestant → Jewish | Religion | 15.000 | 21.000 | 0.300 |
| Jewish → Protestant | Religion | 10.000 | 23.000 | 0.289 |
| Muslim → Protestant | Religion | 10.000 | 23.000 | 0.176 |

Table 17: Steerability experiments across layers 2–30. $\delta$ denotes the steering strength. "Num Target Features" is the number of features used for the target group. "Avg. Improvement" averages 2-option and 3-option improvements.

| Experiment | Attribute | $\delta$ | Num Target Features | Avg. Improvement |
|---|---|---|---|---|
| Less than high school → College graduate/some postgrad | Education | 5.000 | 213.000 | 0.931 |
| Less than $30,000 → $100,000 or more | Income | 5.000 | 252.000 | 0.927 |
| $100,000 or more → Less than $30,000 | Income | 5.000 | 214.000 | 0.805 |
| College graduate/some postgrad → Less than high school | Education | 5.000 | 207.000 | 0.803 |
| Black → Asian | Race | 25.000 | 18.000 | 0.780 |
| Northeast → South | Region | 15.000 | 19.000 | 0.718 |
| Protestant → Atheist | Religion | 10.000 | 81.000 | 0.692 |
| Black → White | Race | 25.000 | 17.000 | 0.651 |
| White → Asian | Race | 25.000 | 18.000 | 0.631 |
| Male → Female | Gender | 25.000 | 6.000 | 0.543 |
| Protestant → Muslim | Religion | 25.000 | 54.000 | 0.520 |
| South → Northeast | Region | 10.000 | 12.000 | 0.498 |
| Asian → White | Race | 15.000 | 17.000 | 0.477 |
| Protestant → Jewish | Religion | 25.000 | 44.000 | 0.374 |
| Atheist → Protestant | Religion | 5.000 | 69.000 | 0.329 |
| Jewish → Protestant | Religion | 5.000 | 69.000 | 0.290 |
| Muslim → Protestant | Religion | 5.000 | 69.000 | 0.203 |

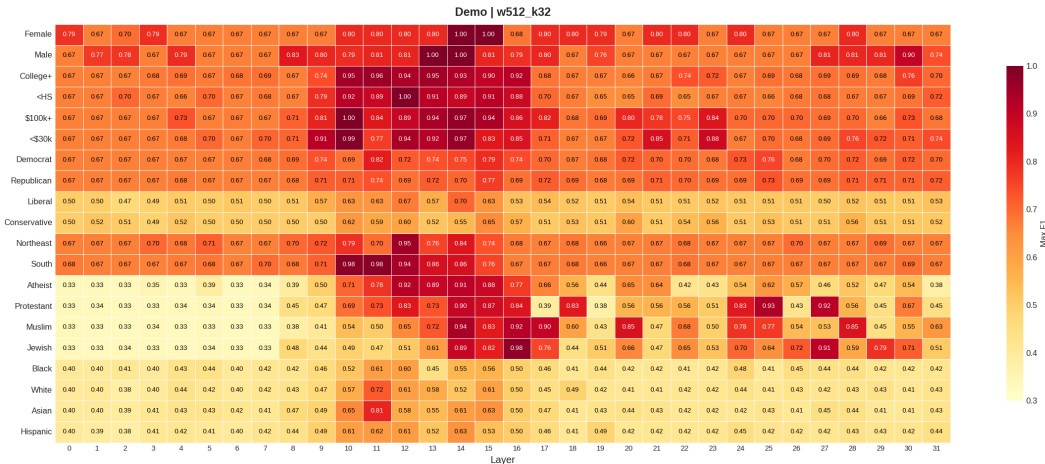

Figure 8: Maximum F1 value achieved by SAE features per demographic group and layer using SAEs with $M_{\text{SAE}} = 512$ and $K_{\text{SAE}} = 32$, using the *demographic token*.

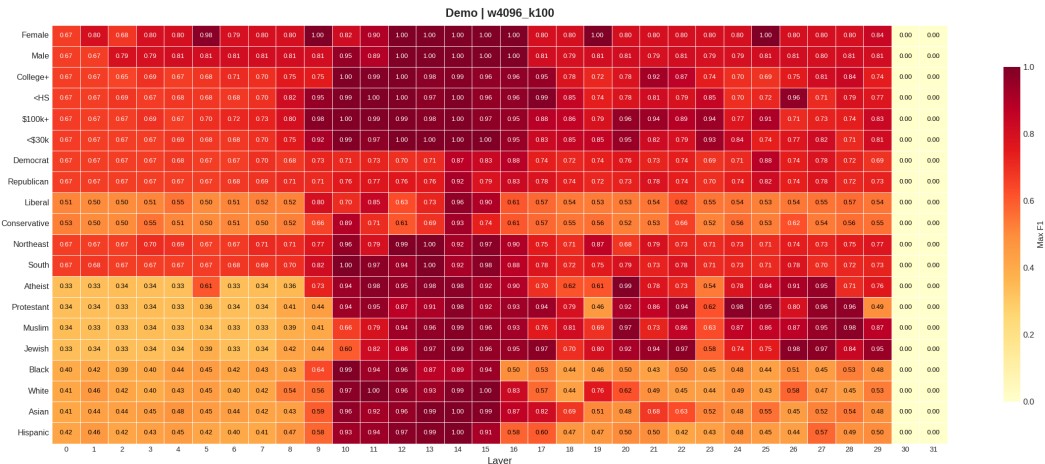

Figure 9: Maximum F1 value achieved by SAE features per demographic group and layer using SAEs with $M_{\text{SAE}} = 4096$ and $K_{\text{SAE}} = 100$, using the *demographic token*.

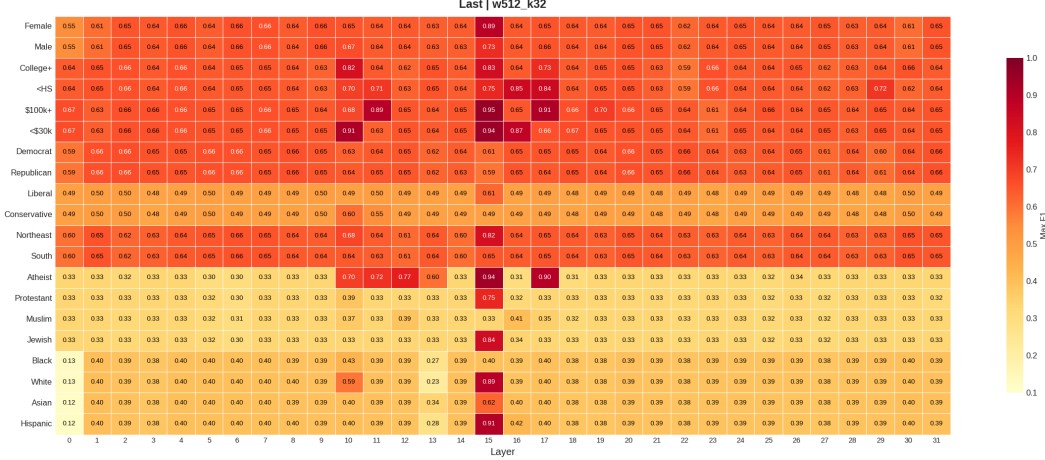

Figure 10: Maximum F1 value achieved by SAE features per demographic group and layer using SAEs with $M_{\text{SAE}} = 512$ and $K_{\text{SAE}} = 32$, using the *last token*.

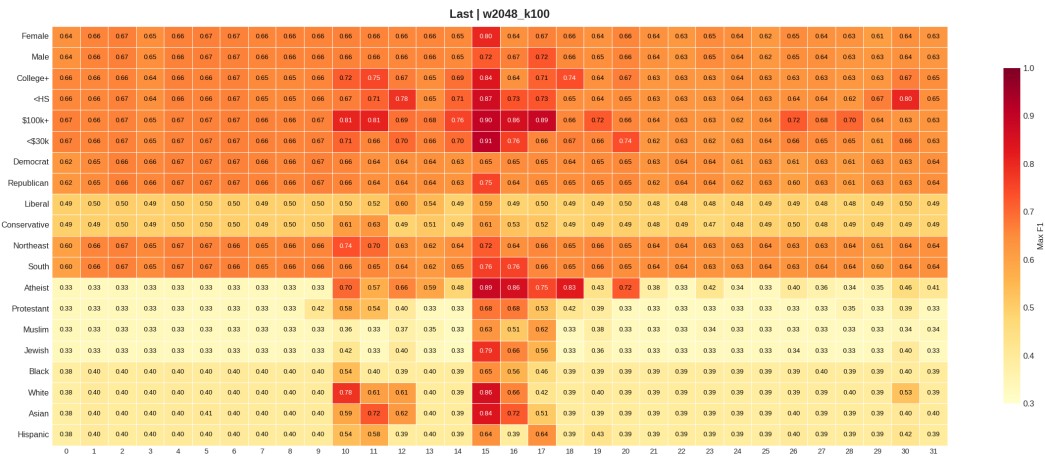

Figure 11: Maximum F1 value achieved by SAE features per demographic group and layer using SAEs with $M_{\text{SAE}} = 2048$ and $K_{\text{SAE}} = 100$, using the *last token*.

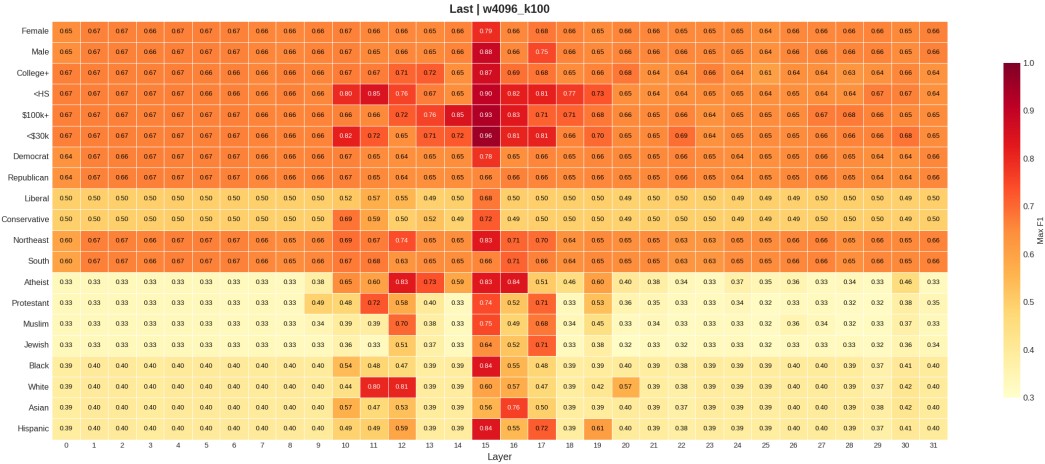

Figure 12: Maximum F1 value achieved by SAE features per demographic group and layer using SAEs with $M_{\text{SAE}} = 4096$ and $K_{\text{SAE}} = 100$, using the *last token*.

