# OpenReview forum: "What Do Large Language Models Know About Opinions?"
_ICLR.cc/2026/Conference — ICLR 2026 Poster_

### Official Review · Reviewer_XXcw · 2025-11-01

**Soundness:** 4
**Presentation:** 4
**Contribution:** 3
**Rating:** 8
**Confidence:** 4

**Summary:**

This paper investigates the internal representation of human opinions in LLMs, studying how these models align with human values and learn during training. Unlike previous studies that focus on output-based evaluations, the authors probe internal model states across 22 demographic groups and a broad range of topics. They find that LLMs possess significantly more accurate internal knowledge of opinions than what is reflected in their outputs, achieving up to 59% better alignment with human responses at a fraction of the computational cost of fine-tuning. The study identifies the emergence of opinion knowledge in middle layers and attributes output discrepancies to final unembedding layers. Using sparse autoencoders, the authors trace opinion-related features to specific attention heads linked to demographic distinctions.

This is a clearly written paper with sound experiments and clear contribution points.

**Strengths:**

- This work addresses the important topic of how investigating internal representations of human opinions across 22 demographic groups.
- Though methodologies used, such as linear or multinomial probing, aren’t original, they were appropriately applied to derive results that are reliable.
- The insights gained through the experiments, such as knowledge of opinions emerge in the middle layers, are useful for the research community.
- The writing is clear.

**Weaknesses:**

- The analysis is conducted on US survey data only.

**Questions:**

- Given the emphasis on the need for pluralistic AI systems, why were the experiments conducted on US survey data only?

---

> ### Author Response · Authors · 2025-11-22
> **Author Response to XXcw**
>
> ## Overview
> We thank the reviewer for the positive and helpful review! We are glad to see that the reviewer appreciated the importance of our topic, soundness of our methods, interesting insights, and clear writing. To address the reviewer’s concern about the analysis only being conducted on US data, we have conducted new experiments on the GlobalOpinionQA dataset, which contains cross-national survey data with responses from countries around the globe.
>
> ## Generalization to non-US contexts
> Thanks, this is a good point! To test whether our results generalize beyond a US context, we have conducted a new set of experiments with the GlobalOpinionQA dataset (Durmus et al, 2023). This dataset contains questions from Pew Global Attitudes Survey and the World Values Survey (WVS), with responses from countries around the globe. Now, instead of predicting the opinions of different demographic groups within the US, we aim to predict the opinions of different countries on cross-national surveys. We keep the top 6 non-US countries in GlobalOpinionQA and fit a probe to predict their opinion distributions, using the same probing methods that we used for the US datasets. We find that our results replicate: first, there is a large improvement from prompting to probing, **with a 44% reduction in KL on average over countries**:
> | Country | Prompt KL | Probe KL | Improvement |
> |---------|-----------|----------|-------------|
> | Germany | 0.207     | 0.095    | 0.54        |
> | France  | 0.211     | 0.095    | 0.55        |
> | Britain | 0.178     | 0.104    | 0.41        |
> | Spain   | 0.198     | 0.108    | 0.45        |
> | Italy   | 0.168     | 0.099    | 0.41        |
> | Poland  | 0.139     | 0.102    | 0.27        |
>
> Second, we see a very similar layer-wise trend, with knowledge emerging rapidly in the middle layers and no loss of knowledge before the final layer, implying that the unembeddings are the source of the discrepancy from probing to prompting.

---

> > ### Comment · Reviewer_XXcw · 2025-11-25
> >
> > Thanks for the response. I'm keeping the score as is.

---

### Official Review · Reviewer_vuiZ · 2025-11-01

**Soundness:** 3
**Presentation:** 3
**Contribution:** 2
**Rating:** 4
**Confidence:** 4

**Summary:**

The paper studies what LLMs “know” about human opinions beyond their surface outputs. Using OpinionQA and SubPOP (US public-opinion surveys) across 22 demographic groups, the authors probe residual-stream activations layer-wise (multinomial logistic vs. MLP probes) to predict full answer distributions to survey questions. Key findings: (i) probes extract substantially more opinion knowledge than next-token probabilities (≈50–60% lower KL), (ii) this knowledge emerges sharply in middle layers (≈layers 10–15), and (iii) the unembedding layer explains much of the gap between internal knowledge and outputs; finetuning only the final linear head recovers probe-level gains and achieves a large fraction of LoRA’s improvements with far fewer parameters. They further train SAEs over attention-head activations and claim group-selective features concentrated in middle layers.

**Strengths:**

- Clear research question & careful operationalization. Evaluating distributional alignment (KL over options) rather than single labels is appropriate for opinions and more informative than argmax accuracy.

- Layerwise analysis identifies where knowledge concentrates. The middle-layer emergence is consistent and useful for targeted interventions.

**Weaknesses:**

I am unconvinced by the novelty and significance of this work. I personally found that the paper largely re-applies established probing methodology to a new label space (opinion/poll distributions) and reports patterns that are unsurprising in light of prior probing literature.

- Major Concerns

   - Incremental Contribution: Most findings (e.g., information concentration in mid layers; linear vs. MLP probe behavior) closely mirror prior probing results.

   - Last-Layer Adaptation Claim: The observation that finetuning or adapting only the output head (unembedding/last layer) recovers most gains is well known from adaptation/calibration lines of work. Presenting this as a key result does not constitute novelty.

   - Probe Design/Analysis: The linear–MLP comparison and the claim that hidden states encode answer distributions are not surprising; no new probing technique, control, or causal intervention is introduced to move beyond correlational readouts.

- Target Definition: Finally, this task predicts polling distributions (as in OpinionQA), which should not be conflated with underlying human opinions. The paper should be explicit about this limitation and avoid over-claiming.

**Questions:**

See above.

---

> ### Author Response · Authors · 2025-11-22
> **Author Response to vuiZ (1/2)**
>
> ## Overview
> We thank the reviewer for their helpful comments. We have conducted several new experiments to address the reviewer's concerns, including new steerability experiments to test whether the SAE features we found have causal implications.
>
> &nbsp;
> ## Major Concerns: Lack of Causal Intervention
> **New steerability experiments.** In response to the reviewer’s concern that no causal intervention is introduced, we have conducted new steerability experiments testing whether the group-specific SAE features we found can be used as causal interventions. Previously, we found that for each of the 22 demographic groups, there are a few distinct SAE features (trained on the model’s attention head activations) that selectively fire when that group is in the prompt. Now, we test whether these features have a causal effect on the LLM’s downstream output. Specifically, we take one base group and one source group and selectively turn off the SAE features of the base group while turning on and magnifying the SAE features of the source group, and then run the LLM forward with the SAE’s reconstructed activations. **We show that our intervention has a causal effect on the LLM’s output** (i.e., predicted opinions from the final residual stream), shifting it much closer to the source group and away from the base group.
>
> We run our steerability experiments on the Democrat and Republican groups in both directions, using Democrats as the source and Republicans as the base, and vice versa. We report the results in the tables below, where the Mean KL reduced and Median KL reduced indicate the mean and median reduction over questions in KL(source, base) after the intervention, and we threshold on questions with at least some divergence between source and base so that the intervention is meaningful. **We find a strong effect with median reductions of 75-92%, demonstrating the efficacy of our causal interventions.**
>
> ------------------------------
>
> **Steering Republican (base) to Democrat (source):**
>
> | Min KL(source, base) | Questions with positive intervention effect | Mean KL reduced | Median KL reduced |
> |-----------------------------|---------------------------------|-----------------|-------------------|
> | 0.100                        | 26/26                           | 0.83            | 0.92              |
> | 0.075                       | 44/46                           | 0.72            | 0.89              |
> | 0.050                        | 90/97                           | 0.62            | 0.75              |
>
> **Steering Democrat (base) to Republican (source):**
>
> | Min KL(source, base) | Questions with positive intervention effect | Mean KL reduced | Median KL reduced |
> |-----------------------------|---------------------------------|-----------------|-------------------|
> | 0.100                        | 23/26                           | 0.62            | 0.81              |
> | 0.075                       | 41/46                           | 0.58            | 0.79              |
> | 0.050                        | 84/97                           | 0.51            | 0.75              |
>
> ------------------------------
>
> **Magnitude Sensitivity Analysis.** We also analyze the relationship between the magnitude we use to steer a group and its effect. We observe that as we increase the magnitude of the features associated with the source group, e.g., Democrat, the model produces more Democrat-sounding results and vice versa for features associated with Republicans.
>
> Below is an example that illustrates the effect of varying the magnitudes of the source feature on the model’s output when steering from Republican to Democrat. We show intervention results for $m \in \\{1, 10, 50, 75 \\} $:
>
> **Question.** Thinking about the level of economic inequality in the country these days, would you say there is
>
> **Answer options:**
>
> - **A.** Too much economic inequality
> - **B.** Too little economic inequality
> - **C.** About the right amount of economic inequality
>
> | Type                      | A prob. | B prob. | C prob. |
> |---------------------------|:-------:|:-------:|:-------:|
> | Democrat (source)         | 0.71| 0.08| 0.21|
> | Republican (base)         | 0.45| 0.12| 0.43|
> | Intervened, \\( m = 1 \\) | 0.48| 0.13| 0.39|
> | Intervened, \\( m = 10 \\)| 0.61| 0.15| 0.24|
> | Intervened, \\( m = 50 \\)| 0.75| 0.11| 0.14|
> | Intervened, \\( m = 75 \\)| 0.89| 0.04| 0.07|

---

> ### Author Response · Authors · 2025-11-22
> **Author Response to vuiZ (2/2)**
>
> ## Major Concerns: Incremental Contribution
>
> **Novelty.** Our work develops a probing and steering framework for a new domain, public opinions. This new domain necessitates several methodological advancements:
> * Most existing works in steerability use steering vectors, which can only steer towards two opposite directions on a single line (e.g., true vs false). In our case, we have 22 different demographic groups that do not lie on a single axis. To support this, we go beyond one-dimensional steering vectors and train an SAE on the attention head activations. A key advantage of our approach is that we can train the SAE once, then steer the model towards the predicted opinions of any of the 22 groups, by magnifying the SAE features corresponding to each group.
> * Our approach is also unusual since most works that train an SAE on model activations train them on the residual stream or MLPs, not the attention heads. Kissane et al (2024) are the first to train SAEs only on attention heads, but they only do this for models up to 2B parameters in size and focus mainly on developing feature attribution methods. **To our knowledge, our work is the first to apply this method to a high-impact domain**, showing that meaningful, causal features can be recovered corresponding to group-specific opinions, and helps to shape this new literature advocating for training SAEs on attention heads.
> * Our setting is also unusual since most probing and steering work focuses on concepts that can be represented by a scalar value, e.g., truthfulness, space and time. In contrast, we are testing the model’s capability to predict opinion distributions. As noted by the reviewer, this requires **innovation in the probe and evaluation**, by developing a multinomial logistic regression probe and using Kullback-Leibler (KL) divergence to measure model fidelity.
>
> Beyond these advancements, our main contribution is in using these methods to study a new and important problem: understanding LLMs’ knowledge of human opinions, which has broad implications for human-AI alignment, political bias in LLMs, and applications of LLMs for social simulation. While many works have tried to assess LLMs’ knowledge of opinions from their outputs, to our knowledge **our work is the first to assess LLMs’ internal knowledge of opinions with probing**. We show that LLMs contain far more knowledge of opinions than revealed by their outputs, isolate unembeddings as the source of the discrepancy, and identify SAE features from attention heads that correspond to different demographic groups. We also demonstrate the practical implications of these findings: our probes **recover over 70% of the improvement from fine-tuning**, providing a lightweight alternative, and the SAE features we found can be used as causal interventions. These findings provide far greater insights of LLMs’ knowledge in this important domain and guide how we should improve human-AI alignment, reduce political bias, and build more accurate and efficient simulations.
>
> &nbsp;
>
> ## Target Definition
> Thanks, this is a good point, and **we have added this limitation to our Discussion section**. Here we refer to answer distributions from public opinion surveys as “opinions”, **following the convention of other work** in the literature that refers to public opinion surveys as opinions, such as “Whose Opinions Do Language Models Reflect?” (Santurkar et al., ICML 2023), “Towards Measuring the Representation of Subjective Global Opinions in Language Models” (Durmus et al., arXiv 2023), or “Aligning language models to user opinions” (Hwang et al., EMNLP 2023). However, we agree that public opinion surveys may not capture all underlying human opinions, such as opinions on personal issues or opinions that cannot be expressed in multiple choice answers.

---

### Official Review · Reviewer_9ET2 · 2025-11-01

**Soundness:** 2
**Presentation:** 3
**Contribution:** 2
**Rating:** 4
**Confidence:** 3

**Summary:**

This paper investigates what large language models (LLMs) internally know about human opinions and how such knowledge is represented in their layers. Using two large-scale U.S. opinion datasets, OpinionQA and SubPOP, the authors probe the residual streams of models such as Llama-3.1-8B, Mistral-7B, and Vicuna-7B to assess alignment with real human opinion distributions across 22 demographic groups. They find that LLMs encode substantially more knowledge about opinions than is evident from their next-token outputs, with probing achieving over 50% lower KL divergence compared to direct prompting. The study contributes to understanding latent representations of social knowledge in LLMs and suggests computationally efficient methods for probing or fine-tuning models for opinion prediction and value alignment.

**Strengths:**

1. Novel methodology and insight
The paper presents a compelling approach to measuring LLMs’ internal representations of opinions rather than relying on surface-level outputs. This perspective offers new insights into how LLMs encode multidimensional social information.

2. Strong empirical and interpretability contribution
By combining probing with sparse autoencoders, the study identifies where and how demographic and opinion-related features emerge in model layers. This combination of layer-wise probing and interpretability analysis is novel and methodologically sound.

3. Computational efficiency and practical implications
Demonstrating that probing achieves comparable gains to fine-tuning at a fraction of the cost provides a practical contribution relevant to alignment, social modeling, and interpretability.

**Weaknesses:**

1. Limited generalizability
The datasets used (OpinionQA and SubPOP) are limited to U.S. opinions. The authors acknowledge this, but cross-cultural validation would be critical to claim broader generalization of “LLMs’ knowledge of opinions.”

2. Unclear link between internal knowledge and actual reasoning
While the paper shows that LLMs encode information internally, it is less clear whether this knowledge can be effectively surfaced during generation. The distinction between “knowing” and “using” opinions could be discussed more deeply.

3. Interpretability conclusions may overreach
Although the use of sparse autoencoders reveals correlations between attention heads and demographic groups, the causal interpretation of these findings remains speculative. Prior work has shown that attention heads often exhibit polysemantic behavior and that apparent interpretability can arise from correlations rather than direct causal encoding (Kissane et al., 2024; O’Neill & Bui, 2024). Similarly, ablation studies such as Baan et al. (2019) demonstrate that removing seemingly interpretable heads often has limited impact on model performance, suggesting that correlation of head activity with a concept does not necessarily imply that the head causally encodes it. The paper would benefit from acknowledging these interpretability limitations and framing its conclusions accordingly.

**Questions:**

1. Could the authors elaborate on how “internal knowledge” might be made actionable? For example, can probing insights be used to steer or align model outputs at inference time?

2. How robust are the findings to prompt variations or to LLMs trained on non-English or cross-cultural corpora?

3. How does the framework compare to retrieval-augmented or social simulation approaches that integrate external evidence (e.g., Group Preference Optimization: Few-Shot Alignment of Large Language Models)?

---

> ### Author Response · Authors · 2025-11-22
> **Author Response to 9ET2 (1/2)**
>
> ## Overview
> We thank the reviewer for the helpful comments. We have conducted several new experiments to address the reviewer’s concerns, including new experiments with GlobalOpinionQA to test whether our results generalize to non-US contexts and new steerability experiments to test whether the SAE features we found have causal implications.
>
> ## W1, Q2: Robustness to Prompt Variations and Non-US Context
>
> **Generalization to non-US contexts.** Thanks, this is a good point! To test whether our results generalize beyond a US context, we have conducted a new set of experiments with the GlobalOpinionQA dataset (Durmus et al, 2023). This dataset contains questions from Pew Global Attitudes Survey and the World Values Survey (WVS), with responses from countries around the globe. Now, instead of predicting the opinions of different demographic groups within the US, we aim to predict the opinions of different countries on cross-national surveys. We keep the top 6 non-US countries in GlobalOpinionQA and fit a probe to predict their opinion distributions, using the same probing methods that we used for the US datasets. We find that our results replicate: first, there is a large improvement from prompting to probing, **with a 44% reduction in KL on average over countries**:
> | Country | Prompt KL | Probe KL | Improvement |
> |---------|-----------|----------|-------------|
> | Germany | 0.207     | 0.095    | 0.54        |
> | France  | 0.211     | 0.095    | 0.55        |
> | Britain | 0.178     | 0.104    | 0.41        |
> | Spain   | 0.198     | 0.108    | 0.45        |
> | Italy   | 0.168     | 0.099    | 0.41        |
> | Poland  | 0.139     | 0.102    | 0.27        |
>
> Second, we see a very similar layer-wise trend, with knowledge emerging rapidly in the middle layers and no loss of knowledge before the final layer, implying that the unembeddings are the source of the discrepancy from probing to prompting.
>
> &nbsp;
>
> **Robustness to prompt variations.** Our results are robust to prompt variations. We try three different prompt variations that have been used in prior work (Santurkar et al., 2023; Suh et al., 2025). These include our default QA format, which conditions the model on the group in a multiple choice question-answering format (Figure 1a); PORTRAY, which instructs the model to answer as if it is a person from that group; and BIO, which describes the group as a first-person biography (see full prompts in Appendix Table 2). Using Llama-3.1-8B, for each format we extract next-token probabilities and train probes on the residual stream per layer. **We find that the results are highly consistent across the different prompting formats**: for 2-option questions, the average KL divergence over groups for next-token probabilities is 0.169 for QA, 0.163 for PORTRAY, and 0.152 for BIO, and the relative improvement of the best-performing probe over the next-token probabilities is 58.6%, 62.6%, and 57.9%, respectively. For 3-option questions, the average KL divergence over groups for next-token probabilities is 0.202 for QA, 0.198 for PORTRAY, and 0.20 for BIO, and the relative improvements from the best-performing probe is 44.6%, 40.9%, and 42.5%, respectively.
>
> &nbsp;
>
> ## Q3: Comparison to Other Approaches with External Evidence
> As described by the reviewer, other approaches, such as group preference optimization (GPO), aim to improve the model’s predictions of group preferences or opinions by providing external evidence. In contrast, our framework aims to make more out of what the model already knows, instead of providing it with external information, such as few-shot examples of responses from that group (like GPO). The improvement from prompting to our probes is that our probes found information already contained in the model’s internal layers, but were not being surfaced during generation. We have added this comparison to our Related Work section.
>
> In terms of how these approaches compare in terms of performance, **GPO falls outside the scope of our paper, since the goal of our paper is to better understand LLM’s internal knowledge of opinions**, instead of seeing what they could do with external evidence. However, we do know that our probes achieve over 70% of the improvement from fine-tuning on many examples from each group, so we expect our probes should achieve at least 70% of the improvement from providing few-shot examples like GPO.

---

> ### Author Response · Authors · 2025-11-22
> **Author Response to 9ET2 (2/2)**
>
> ## W2, W3, Q1: Making Knowledge Actionable through Steering
> Thanks, we agree that it is valuable to show how the internal knowledge we found can be made actionable and to test whether the SAE features we found are causal. We have conducted a new set of steerability experiments to explore these questions.
>
> Previously, we found that for each of the 22 demographic groups, there are a few distinct SAE features (trained on the model’s attention head activations) that selectively fire when that group is in the prompt. Now, we test whether these features have a causal effect on the LLM’s downstream output. Specifically, we take one base group and one source group and selectively turn off the SAE features of the base group while turning on and magnifying the SAE features of the source group, and then run the LLM forward with the SAE’s reconstructed activations. **We show that our intervention has a causal effect on the LLM’s output** (i.e., predicted opinions from the final residual stream), shifting it much closer to the source group and away from the base group and demonstrate how the **internal knowledge that we found can be made actionable**.
>
> We run our steerability experiments on the Democrat and Republican groups in both directions, using Democrats as the source and Republicans as the base, and vice versa. We report the results in the tables below, where the Mean KL reduced and Median KL reduced indicate the mean and median reduction over questions in KL(source, base) after the intervention, and we threshold on questions with at least some divergence between source and base so that the intervention is meaningful. **We find a strong effect with median reductions of 75-92%, demonstrating the efficacy of our causal interventions.**
>
> ------------------------------
>
> **Steering Republican (base) to Democrat (source):**
>
> | Min KL(source, base) | Questions with positive intervention effect | Mean KL reduced | Median KL reduced |
> |-----------------------------|---------------------------------|-----------------|-------------------|
> | 0.100                        | 26/26                           | 0.83            | 0.92              |
> | 0.075                       | 44/46                           | 0.72            | 0.89              |
> | 0.050                        | 90/97                           | 0.62            | 0.75              |
>
> **Steering Democrat (base) to Republican (source):**
>
> | Min KL(source, base) | Questions with positive intervention effect | Mean KL reduced | Median KL reduced |
> |-----------------------------|---------------------------------|-----------------|-------------------|
> | 0.100                        | 23/26                           | 0.62            | 0.81              |
> | 0.075                       | 41/46                           | 0.58            | 0.79              |
> | 0.050                        | 84/97                           | 0.51            | 0.75              |
>
> ------------------------------
>
> **Magnitude Sensitivity Analysis.** We also analyze the relationship between the magnitude we use to steer a group and its effect. We observe that as we increase the magnitude of the features associated with the source group, e.g., Democrat, the model produces more Democrat-sounding results and vice versa for features associated with Republicans.
>
> Below is an example that illustrates the effect of varying the magnitudes of the source feature on the model’s output when steering from Republican to Democrat. We show intervention results for $m \in \\{1, 10, 50, 75 \\} $:
>
> **Question.** Thinking about the level of economic inequality in the country these days, would you say there is
>
> **Answer options:**
>
> - **A.** Too much economic inequality
> - **B.** Too little economic inequality
> - **C.** About the right amount of economic inequality
>
> | Type                      | A prob. | B prob. | C prob. |
> |---------------------------|:-------:|:-------:|:-------:|
> | Democrat (source)         | 0.71| 0.08| 0.21|
> | Republican (base)         | 0.45| 0.12| 0.43|
> | Intervened, \\( m = 1 \\) | 0.48| 0.13| 0.39|
> | Intervened, \\( m = 10 \\)| 0.61| 0.15| 0.24|
> | Intervened, \\( m = 50 \\)| 0.75| 0.11| 0.14|
> | Intervened, \\( m = 75 \\)| 0.89| 0.04| 0.07|
>
> ------------------------------
>
> At the same time, while we are currently conducting steerability experiments for other groups, we agree with the reviewer that correlations between attention heads or SAE features and concepts do not imply that those units causally encode the concepts. In the camera-ready version, we will revise section 4 on the results of our SAE to (1) acknowledge these limitations between correlation and causality, (2) present our new steerability results, and (3) explicitly frame the features and heads found by the SAE as **useful testable hypotheses** about where group-specific information may be represented.

---

### Official Review · Reviewer_Afma · 2025-11-01

**Soundness:** 3
**Presentation:** 2
**Contribution:** 2
**Rating:** 4
**Confidence:** 3

**Summary:**

This paper investigates opinion-related knowledge encoded within large language models (LLMs). To identify latent knowledge in LLMs, the authors employ probing and sparse autoencoder techniques. Specifically, they extract predicted probabilities from a trained probe given opinion-eliciting prompts and compare these against next-token probabilities, measuring divergence from ground-truth human opinion distributions. The experimental evaluation utilizes two opinion datasets containing demographic information. Results demonstrate that probing-based predictions align more closely with ground-truth distributions than standard LLM outputs. Through comprehensive layer-wise and option-wise analyses, the authors reveal that middle layers encode the richest opinion-related knowledge. The authors conclude that LLMs encode substantially more internal knowledge about opinions than their generated outputs suggest.

**Strengths:**

The primary strength of this work lies in its revealing findings about hidden knowledge in LLMs for opinion generation tasks. The results indicate that LLMs possess greater capacity to generate demographically appropriate opinions than their outputs reflect, suggesting interesting limitations in the generation process. The methodology, while conceptually straightforward, is sound and reproducible, providing a template that other researchers can adapt for similar investigations.

**Weaknesses:**

The paper would benefit from deeper analysis in several areas. The observation in line 321 “This suggests that the drop from probe to prompting arises at the unembedding stage” is particularly intriguing but underexplored. While the authors provide additional investigation, the presentation consists primarily of reported numbers without sufficient interpretation. Visualizing these results in a format similar to Figure 4 would help highlight the pattern where final layers underperform middle layers. Without such emphasis and thorough analysis, it remains unclear how this work's findings differ substantively from prior research examining layer-specific roles in other tasks, such as multilingualism [1].

[1] Zhao, Yiran, Wenxuan Zhang, Guizhen Chen, Kenji Kawaguchi, and Lidong Bing. "How do large language models handle multilingualism?" Advances in Neural Information Processing Systems 37 (2024): 15296-15319.

**Questions:**

The nature of the captured knowledge requires clarification. The authors frame their research question as "what do LLMs know about human opinions?" and examine differences across demographic groups. However, it is ambiguous whether the results reveal knowledge specifically about demographics, opinions, or their interaction. To strengthen the claims, the authors should conduct control experiments with non-opinion tasks (e.g., cultural knowledge or mathematical reasoning) using the same methodology. Do similar layer-wise patterns emerge? Are the demographic effects opinion-specific or more general? Such investigations are essential to establish whether the findings are unique to opinion modeling or reflect broader properties of LLM internal representations.

---

> ### Author Response · Authors · 2025-11-22
> **Author Response to Afma (1/3)**
>
> ## Overview
>
> We thank the reviewer for the helpful comments. We have conducted several new experiments to address the reviewer’s concerns, deepening the analyses in the paper and clarifying the nature of the captured knowledge.
>
> ## W1: Deeper Analysis
> We agree the paper would benefit from deeper analysis and interpretation of the reported numbers. First, to respond to the reviewer’s specific suggestions:
>
> **Unembedding implications:** As noted by the reviewer, our finding that the drop from probe to prompting arises at the unembedding stage is intriguing. We have added further analysis of the implications. First, this result indicates where the model is losing information. Luckily, instead of the model gradually losing information throughout the residual stream, which would be harder to recover, it is abruptly losing it in the unembedding layer. Second, as a practical matter, this means that we can recover the knowledge by only changing the unembedding layer. We demonstrate this by fine-tuning only the unembedding layer, showing that (1) this resolves the gap between the probe and prompting and (2) **fine-tuning only the unembedding layer achieves 75-85% of the improvement from LoRA fine-tuning**, providing a lightweight alternative to fine-tuning the entire model.
>
> **Zhao et al:** We have added the Zhao et al (2024) paper to our Related Work section and explained how the works relate to each other. Zhao et al study the LLM’s multilingual workflow over layers: beginning by understanding the query in non-English, converting the query into English, employing English for reasoning in the middle layers, then generating language aligned with the original language in the final layers. This is related to our work since we are also trying to understand in which layers knowledge of opinions emerges and is lost in the LLM. However, our work is different in a number of ways, including (1) we study the model’s knowledge and processing of human opinions, instead of multilingualism, (2) we demonstrate a large discrepancy between prompting and probing in this domain, and (3) we train an SAE to identify features in the attention heads that are responsible for different demographic groups.
>
> We have also added deeper analysis in several other parts of the paper:
>
> **The nature of knowledge in the residual stream**: We have extended our discussion of the probing results. Previously, we found that the logistic regression probe (a generalized linear model) performs as well as an MLP probe, which is interesting since it’s hard to see how the opinions of 22 demographic groups that do not lie on a single line can be linear. However, it is important to distinguish between opinions as the answer distribution vs. as the features involved in building the opinions (e.g., the views of the group or the issues involved in the question). We hypothesize that by the point we are looking at the most predictive residual stream layer, **the knowledge of opinions encoded in the residual stream is the answer distribution**, which can be represented linearly, and this is indeed what the probe is trained to predict. We further test this hypothesis by comparing the performance of probes when using one probe per group versus one probe for all groups. If there is a consistent representation of answer distribution that emerges in the residual stream after all group and question information is added in, we should see that a single probe for all groups suffices. This is what we see for Llama-3.1-8B, where we find that the probe for all groups performs almost exactly the same as the probe per group, with an average KL difference of 0.001 for 2-option and 0.004 for 3-option questions (Tables 5-6). Thus, what is linearly encoded in the residual stream is the answer distribution after features such as the group and question information have been added in.
>
> **What is learned during fine-tuning**: We have added further discussion of our fine-tuning results. Previously, we found that fine-tuning greatly improves the LLM’s ability to predict opinions, but probing achieves over 70% of that improvement while being 278× more parameter efficient. We have added further discussion of the implications. As a practical matter, these results reveal that probes could serve as an effective lightweight alternative to fine-tuning. Furthermore, since probes do not teach the LLM new knowledge—they can only use knowledge that is already in the LLM—these results suggest that **a majority of the fine-tuning improvement is attributable to the LLM learning how to better retrieve and format knowledge that it already had**, as opposed to learning new knowledge, providing evidence for the “Less is More for Alignment” (LIMA) hypothesis (Zhou et al., 2023) in a new domain.

---

> ### Author Response · Authors · 2025-11-22
> **Author Response to Afma (2/3)**
>
> ## W1: Deeper Analysis (cont.)
>
> **Steerability experiments**: We have also conducted a new set of experiments deepening our analysis of the SAE features. Previously, we found that for each of the 22 demographic groups, there are a few distinct SAE features (trained on the model’s attention head activations) that selectively fire when that group is in the prompt. Now, we test whether these features have a causal effect on the LLM’s downstream output. Specifically, we take one base group and one source group and selectively turn off the SAE features of the base group while turning on and magnifying the SAE features of the source group, and then run the LLM forward with the SAE’s reconstructed activations. **We show that our intervention has a causal effect on the LLM’s output** (i.e., predicted opinions from the final residual stream), shifting it much closer to the source group and away from the base group.
>
> We run our steerability experiments on the Democrat and Republican groups in both directions, using Democrats as the source and Republicans as the base, and vice versa. We report the results in the tables below, where the Mean KL reduced and Median KL reduced indicate the mean and median reduction over questions in KL(source, base) after the intervention, and we threshold on questions with at least some divergence between source and base so that the intervention is meaningful. **We find a strong effect with median reductions of 75-92%, demonstrating the efficacy of our causal interventions.**
>
> ------------------------------
>
> **Steering Republican (base) to Democrat (source):**
>
> | Min KL(source, base) | Questions with positive intervention effect | Mean KL reduced | Median KL reduced |
> |-----------------------------|---------------------------------|-----------------|-------------------|
> | 0.100                        | 26/26                           | 0.83            | 0.92              |
> | 0.075                       | 44/46                           | 0.72            | 0.89              |
> | 0.050                        | 90/97                           | 0.62            | 0.75              |
>
> **Steering Democrat (base) to Republican (source):**
>
> | Min KL(source, base) | Questions with positive intervention effect | Mean KL reduced | Median KL reduced |
> |-----------------------------|---------------------------------|-----------------|-------------------|
> | 0.100                        | 23/26                           | 0.62            | 0.81              |
> | 0.075                       | 41/46                           | 0.58            | 0.79              |
> | 0.050                        | 84/97                           | 0.51            | 0.75              |
>
> ------------------------------
>
> **Magnitude Sensitivity Analysis.** We also analyze the relationship between the magnitude we use to steer a group and its effect. We observe that as we increase the magnitude of the features associated with the source group, e.g., Democrat, the model produces more Democrat-sounding results and vice versa for features associated with Republicans.
>
> Below is an example that illustrates the effect of varying the magnitudes of the source feature on the model’s output when steering from Republican to Democrat. We show intervention results for $m \in \\{1, 10, 50, 75 \\} $:
>
> **Question.** Thinking about the level of economic inequality in the country these days, would you say there is
>
> **Answer options:**
>
> - **A.** Too much economic inequality
> - **B.** Too little economic inequality
> - **C.** About the right amount of economic inequality
>
> | Type                      | A prob. | B prob. | C prob. |
> |---------------------------|:-------:|:-------:|:-------:|
> | Democrat (source)         | 0.71| 0.08| 0.21|
> | Republican (base)         | 0.45| 0.12| 0.43|
> | Intervened, \\( m = 1 \\) | 0.48| 0.13| 0.39|
> | Intervened, \\( m = 10 \\)| 0.61| 0.15| 0.24|
> | Intervened, \\( m = 50 \\)| 0.75| 0.11| 0.14|
> | Intervened, \\( m = 75 \\)| 0.89| 0.04| 0.07|

---

> ### Author Response · Authors · 2025-11-25
> **Author Response to Afma (3/3)**
>
> ## Q1: Are the Findings Unique to Demographics and Opinions?
> **Opinions, no demographics.** We have conducted a new set of experiments with the GlobalOpinionQA dataset, which contains country-specific responses to cross-national surveys. So, instead of predicting the opinions of different demographic groups within the US, we aim to predict the opinions of different countries. We conducted this experiment in part due to concerns from other reviewers about the US-centric nature of our original two datasets, but this can also serve as a control experiment of an opinion task without demographics. In this case, we find that our results replicate: first, there is a large improvement from prompting to probing, with a **44% reduction in KL on average over countries**:
>
> | Country | Prompt KL | Probe KL | Improvement |
> |---------|-----------|----------|-------------|
> | Germany | 0.207     | 0.095    | 0.54        |
> | France  | 0.211     | 0.095    | 0.55        |
> | Britain | 0.178     | 0.104    | 0.41        |
> | Spain   | 0.198     | 0.108    | 0.45        |
> | Italy   | 0.168     | 0.099    | 0.41        |
> | Poland  | 0.139     | 0.102    | 0.27        |
>
> Second, we see a very similar layer-wise trend, with knowledge emerging rapidly in the middle layers. So, it appears that our findings about demographics and opinions are not specific to demographics, but we have yet to study whether they are specific to opinions.
>
> **Demographics, no opinions.** Separately, we agree that it would be interesting to explore the opposite, as suggested by the reviewer, i.e., some demographic but non-opinion task. For example, if the model is conditioned on a demographic group within the US, can it correctly predict the group’s distribution over income levels or household consumption, as a more encyclopedic than opinion-based task? Or could it correctly predict their habits and behaviors (e.g., dietary behaviors from NHANES, mobility patterns from GPS data)? We will add an experiment in this direction before the camera-ready version of our paper.

---

### Author Response · Authors · 2025-12-03
**Summary of responses and changes (1/2)**

## Overview
Thanks to all the reviewers for their helpful and constructive reviews. We are glad that **all the reviewers** appreciated the soundness of our experiment design and results. Most of the reviewers **(Afma, 9ET2, XXcw)** found the insights from our findings novel and reproducible with practical applications for the research community.

We received an average score of 5.0 from reviewers, with one 8 (accept, good paper (poster)) from XXcw, who reaffirmed their score during the discussion period, and three 4’s (marginally below the acceptance threshold. But would not mind if paper is accepted), who did not have a chance to respond to our rebuttal. **We strongly believe that if they had been given the opportunity to respond, they would have updated their scores above the acceptance threshold**, given that they already were very close to the threshold and we thoroughly addressed their concerns.
Below, we highlight two new experiments we ran that address the concerns of all four reviewers, then describe in greater point-by-point detail how we responded to each reviewer’s concerns.

## Main New Experiments

**1. Steerability via SAE features.** To address concerns about lacking causal interventions and actionability, we conducted new steerability experiments using SAE features. Previously, we trained SAE features on the attention head activations and found, for each of the 22 demographic groups, that there were SAE features that selectively fired (with F1s above 0.9) when that group was in the prompt. Now, we test whether these features can be used for causal interventions. For a chosen base group and source group (e.g., Democrat as base and Republican as source, or vice versa), we selectively turn off SAE features associated with the base group and turn on/magnify those for the source group, then run the model forward with the SAE’s reconstructed activations. We report mean and median fractional reductions in KL(source, base) across questions above a minimum divergence threshold. **We find strong effects, with median KL reductions of 0.75–0.92, and a clear dependence on intervention magnitude**, showing that the identified internal features can be used to reliably steer the model’s opinion distributions toward a target group. Please see [this response](https://openreview.net/forum?id=kHVzEjThKE&noteId=ZLtxihX99g) for a detailed table.

**This new set of experiments addressed primary concerns of all three reviewers who gave us a score of 4.** Reviewer 9ET2 wanted to understand how “knowledge can be effectively surfaced during generation” and made actionable, and questioned the causal interpretation of our findings; these results show that the SAE features are causal and actionable, allowing us to controllably change the LLM’s outputs. Reviewer vuiZ was concerned that no causal intervention is introduced and questioned the methodological novelty of our work; these experiments demonstrate a causal intervention and propose a new SAE-based method, going beyond steering vectors, which allows us to steer the model towards 22 demographic groups after only training the SAE once. Finally, reviewer Afma wanted to see “deeper analysis” and clarification of the “nature of the captured knowledge”; these experiments deepen our analysis from correlational to causal and clarify the nature of captured knowledge by isolating group-specific features in the attention heads.

**2. Generalization to non-US contexts.** To test whether our findings extend beyond US demographic groups, we ran new experiments on the GlobalOpinionQA dataset (Durmus et al., 2023), which contains cross-national survey questions from Pew Global Attitudes Survey and World Values Survey. Instead of predicting opinions of demographic groups within the US, we now predict opinion distributions over countries. We keep the top 6 non-US countries and fit probes using the same methodology as in the main paper. We find that our main result replicates: probing again substantially outperforms prompting, with an average KL reduction of ≈44% over countries, and we observe a similar layer-wise pattern with knowledge emerging in the middle layers and preserved up to the final layer. Please see [this response](https://openreview.net/forum?id=kHVzEjThKE&noteId=W2Aeclgn5B) for a detailed table.

**This new set of experiments addressed primary concerns of three reviewers**. First, reviewer 9ET2 noted, “the datasets used (OpinionQA and SubPOP) are limited to U.S. opinions”, and reviewer XXcw said, “The analysis is conducted on US survey data only.” Our new results address this shared concern about only testing on US data and demonstrate that our results generalize to non-US opinions. Second, reviewer Afma wanted to understand if our results were unique to "demographics, opinions, or their interaction", and this analysis isolates opinions but not demographic groups (country-level opinions instead), showing that results generalize in this case.

---

> ### Author Response · Authors · 2025-12-03
> **Summary of responses and changes (2/2)**
>
> ## Point-by-point summary
> Below is a summary of the main concerns from each reviewer and how we have carefully addressed them through new experiments and clarifications. Further details of the experiments and clarifications are described in the respective responses under the reviewers' comments.
>
> &nbsp;
>
> **Reviewer Afma**
>
> *Main concerns and questions.*
>
> - **Need for deeper analysis of the results and the nature of knowledge captured by probes.**
>   We added new analyses of SAE features (with causal interventions), a magnitude sensitivity analysis, and further discussed the experiments on what exactly is encoded in the residual stream.
>
> - **Further investigation of the drop from probing to prompting at the unembedding stage.**
>   We presented our targeted fine-tuning experiments on the unembedding layer and compared them to LoRA to localize the gap and quantify how much can be recovered by last-layer adaptation.
>
> - **Clearer differentiation from prior work on layer-specific behavior (e.g., Zhao et al., 2024).**
>   We clarified how our setting, methods, and findings differ from previous work: new domain (opinions), multi-group distributions, SAE on attention heads with steering, and explicit localization of the prompting–probing gap to the unembedding.
>
> - **Are the findings specific to opinions and demographics?**
>   We added experiments on GlobalOpinionQA, predicting country-level opinion distributions instead of US demographic groups, and showed that our main phenomena replicate.
>
>
> ------------------------------
>
> **Reviewer 9ET2**
>
> *Main concerns and questions.*
>
> - **Limited generalizability beyond US datasets.**
>   We added experiments on GlobalOpinionQA (cross-national survey questions) and showed that our findings generalized, with the same gap between prompting and probing and a similar layer-wise pattern.
>
> - **Unclear link between internal knowledge and generation; how to make it actionable.**
>   We show that probes recover most of the gains of fine-tuning and add new SAE-based steerability experiments that directly steer opinion distributions at inference time.
>
> - **Interpretability and causality concerns around SAE features.**
>   We add targeted causal interventions, and in the camera-ready we explicitly frame SAE features as testable hypotheses rather than definitive causal units.
>
> - **Robustness to prompt variations.**
>   We tested three prompting formats from prior work and found that both base KL and relative improvements from probes are stable across formats.
>
> - **Positioning relative to external-evidence approaches (e.g., GPO).**
>   We clarify that our focus is on internal knowledge (no extra group-specific data) and relate our gains to those from fine-tuning-based methods.
>
> ------------------------------
>
> **Reviewer vuiZ**
>
> *Main concerns and questions.*
>
> - **Incremental contribution and novelty.**
>   We agree that our probing setup may mirror prior work; however, we emphasize the novelty of the domain-specific findings (opinion distributions across many groups), the SAE-on-heads steering framework, and the new causal intervention results in this setting.
>
> - **Probe design and lack of causal intervention.**
>   We added SAE-based steerability experiments and magnitude sensitivity analysis, making the interpretability results more than purely correlational.
>
> - **Target definition: polling distributions vs “true opinions.”**
>   We clarify in the Discussion that we model answer distributions from public opinion surveys, following prior work, and explicitly acknowledge that these do not capture all underlying human opinions.
>
> ------------------------------
>
> **Reviewer XXcw**
>
> *Main concerns and questions.*
>
> - **Analysis only conducted on US survey data.**
>   We added experiments on GlobalOpinionQA (cross-national survey questions) and showed that our findings generalized, with the same gap between prompting and probing and a similar layer-wise pattern.

---

### Meta-Review · Area_Chair_DFZJ · 2026-01-11

**Summary:**

The paper investigates the internal knowledge of large language models (LLMs) about human opinions across 22 demographic groups, using probing rather than surface-level next-token outputs. It finds that LLMs encode substantially more opinion knowledge internally—achieving 50–59% lower KL divergence from human survey distributions than prompting—without fine-tuning and at 278× lower computational cost. This internal knowledge emerges in middle layers and is lost primarily at the unembedding stage. Using sparse autoencoders (SAEs), the authors identify attention-head features specific to demographic groups and demonstrate causal steerability by intervening on these features. All reviewers recognize the strong motivation, methodological rigor, and novel insights into LLM alignment. Concerns about US-centric data, correlational (vs. causal) interpretation, and incremental novelty were largely addressed through new experiments on GlobalOpinionQA and SAE-based steering.

**Reviewer Concerns:**

Addressed by rebuttal:

US-only data: Authors added experiments on GlobalOpinionQA (cross-national Pew/World Values Survey data), showing the same prompting–probing gap and layer-wise pattern for country-level opinion prediction.
Correlation vs. causality: New SAE-based steerability experiments show that activating/deactivating group-specific features causally shifts model outputs toward target demographics (median KL reduction: 75–92%).
Actionability: Demonstrated that probes recover >70% of fine-tuning gains with negligible compute, and SAE features enable real-time inference-time steering.
Unembedding drop: Confirmed via targeted fine-tuning that adapting only the unembedding recovers most probing gains, localizing the bottleneck.
Robustness to prompts: Tested three prompt formats; results are stable across variations.
Still outstanding (minor):

Ethical framing: While not flagged for full ethics review, the paper equates survey responses with “opinions”; authors now clarify this limitation but do not address potential biases in underlying datasets (e.g., underrepresentation).
Generalization beyond opinions: The method’s applicability to non-opinion tasks (e.g., cultural facts) remains untested, though authors plan to explore this.

**Reviewer Scores:**

Afma (initial: 4 – marginally below): Raised concerns about analysis depth and generalizability. Rebuttal added causal steering, cross-national results, and unembedding analysis.

9ET2 (initial: 4 – marginally below): Concerned about US focus, actionability, and interpretability limits. All addressed with GlobalOpinionQA and steering experiments.

vuiZ (initial: 4 – marginally below): Viewed contribution as incremental and lacking causal intervention. New steerability results directly counter this.

XXcw (initial: 8 – accept): Already strongly positive; appreciated new cross-cultural results.

---

### Decision · Program_Chairs · 2026-01-26

Accept (Poster)